# The (non)effect of personalization in climate texts on credibility of climate scientists: A case study on sustainable travel

Anna Leerink[1], Mark Bos[1], Daan Reijnders[2], Erik van Sebille[1,2]

[1]Freudenthal Institute, Utrecht University, Utrecht, 3584CC, Netherlands

[2]Institute for Marine and Atmospheric research, Utrecht University, Utrecht, 3584CC, Netherlands

*Correspondence to*: Erik van Sebille (E.vanSebille@uu.nl)

**Abstract.** How we communicate about climate change affects how others think, feel and act. Therefore, the way climate scientists formulate messages is important. In this study, we assess the effect of personalization, operationalized as writing in a conversational style, as previously done by Ginns and Fraser [2010], and perceived credibility of climate scientists. We exposed one hundred participants aged between 18 and 35 to three conditions of a text on the climate impact of train versus plane travel, with varying degree of personalization, and assessed the outcome in their attitude (specifically interest and opinion) towards sustainable travel, as well as the perceived credibility of the climate scientist who wrote the text. Results show that there is a small effect in the degree of happiness after reading the different texts, but little other effects. Our main conclusion is that, although personalization may be well received by readers, it may not be the best mode to influence the attitudes of readers towards sustainable travel, nor how readers come to perceive climate scientists' credibility.

## 1 Introduction

Climate change, due to anthropogenic carbon emissions, is a major environmental problem. One critical driver for climate action is the public's attitude (specifically interest and opinion): a study suggests that positive attitudes towards climate-related topics lead to higher support for climate action [Cerf et al 2023]. Attitudes can be affected by many factors, including the perceived credibility of information providers when reading climate information [Scott and Willits 1994, Dong *et al* 2018, Bouman *et al* 2021]. In their study, Dong *et al* [2018] found that there is a positive relationship between climate information and action, and that it can be strengthened by the perceived credibility of the information provider. When it is understood how specific textual elements affect the perceived credibility of information providers, this information can be used to optimally strengthen the relationship between climate information and climate action.

One way of appealing to the emotional involvement and happiness with the text is implementing certain textual elements that have emotional appeal [Glaser *et al* 2009]. Indeed, previous research showed that highlighting sadness or hope, or using gain or loss frames, can affect readers' responses [Lu 2016]. There have been numerous studies on how narrative elements (specifically written in a manipulative way, such as done by lobbyists) affect people's responses, and how these elements can improve knowledge acquisition [Norris *et al* 2005, Glaser *et al* 2009, Dahlstrom 2014, Yang and Hobbs 2020]. A combination between expository – purely scientific – and narrative elements is often used to popularize science and stimulate interest [Avraamidou and Osborne 2009]. One such element is personalization, which is here defined as a way of

communicating abstract scientific concepts within a frame of reference, focusing on a particular individual or smaller group of people and exploring their actions and the consequences these uphold [Schiffer and Guerra 2015, Vonk *et al* 2023].

Personalization of expository texts can affect the reading experience, by creating a protagonist that 'explains' the science
[Glaser *et al* 2009]. This protagonist decreases the distance between the reader and the content of the text and can thus urge readers to actively participate in reading, which leads to a feeling of closer proximity [Sangers *et al* 2020]. Additionally, such elements are likely to make the content emotionally more interesting. One way to include personalization in a text is to use direct address, where the writer addresses the reader in the second-person voice with 'you'. Another way is for the writer to explicitly expose themselves as the protagonist, writing in the first-person voice and including opinions in the text.
Multiple studies showed that personalization enhances learning outcomes and understanding [Ginns and Fraser 2010, Mayer 2014, Sangers *et al* 2020].

However, whether personalization also results in an attitude change on climate change is understudied [Cerf *et al* 2023]. Understanding the effect of personalization on the public's attitude can thus inform about the usefulness of personalization in climate mitigation and adaptation. Such insights may help climate communicators decide on their mode of communication
and formulation of their message. This, for example, can help climate communicators write popular scientific translations of highly scientific – expository – research, including the Intergovernmental Panel on Climate Change (IPCC) reports. Therefore, our first research question is:

*RQ1. How does personalization of popular scientific climate texts affect the interest and opinion on climate change of participants?*

Scientists' roles in public dialogues have been discussed by the scientific community persistently [Pielke Jr 2007]. Especially in climate communication, knowing what role to take can be hard [Fischhoff 2007]. Considerations for scientists conducting climate communication can, for example, be the wish to remain neutral, or to reflect objectivity, resulting in a specific type of text that will be very different from one written by scientists considering it is their role to convince or incite the public to action and urge for change. Often, scientists choose to communicate in the role of *pure scientist,* aiming to
provide neutral, unbiased, and fundamental information [Pielke Jr 2007]. Scientists might be worried about their perceived credibility, when choosing another role, such as that of issue advocate. However, communicating in the role of *issue advocate* can make information more comprehensible for a broader audience [Cologna *et al* 2021]. In this role, scientists inform the public of their own preference by explicitly voicing their support for one policy over others [Pielke Jr 2007].

By adding direct address to the reader and by exposing the writer as the protagonist, the role of a scientist in climate texts
may shift from pure scientist to issue advocate. It is, however, not yet known how these types of personalization affect the perceived credibility of a text or the scientist who wrote it. To find out more about this effect, our second research question is:

*RQ2. What is the effect of personalization on the perceived credibility of a popular scientific text and the climate scientist who wrote it?*

A new instrument proposed and tested by Peeters *et al* [2022], and which we will be using here, can serve just that purpose. Their *IMPACTLAB* instrument is a toolbox, specifically designed for science communication, that provides a set of tools to measure the effect of public engagement activities. It also includes a decision tree to choose the most appropriate measurement tool for a particular activity. ~~It's~~It is based on a theoretical framework to measure three features that help evaluate science communication interventions: science capital (what Peeters *et al* [2022] term "output"), emotional memory

("outcome") and long-term effect ("impact"). The science capital of participants is measured to find out how acquainted the public is with science in general. The emotional memory measures which emotions are aroused with the public. Emotions serve as predictors of memory retention, influencing how effectively individuals recall experiences over the long term. Additionally, the effect analysis measures a change in attitude. Within the framework, it is realized that measuring output is relatively straightforward, but that measuring impact can be extremely difficult. The strength of the tool is that it is very

practical and easy to adapt to a wide variety of public engagement activities.

To answer the two research questions, we conducted a randomized online survey experiment in which participants read a popular scientific text and answered questions. Based on a design with three different conditions (i.e., expository, slightly personalized, and highly personalized), both the effect of personalization on the perceived credibility of the climate scientist who wrote it and the effect of personalization on participants' attitude (specifically interest and opinion) toward sustainable

travel were studied. As the basis for the three texts, we used an existing and published online popular science article. In this original text, the carbon emissions of travelling by train are compared to those of flying, while also considering the building of infrastructure.

## 2. Methods

### 2.1 Context

The popular science article was taken from the Klimaathelpdesk.org (KH), a Dutch online platform where society can ask questions about climate change to academic experts. These questions are published along with academic peer-reviewed answers, which include references. Questions that are sent to the KH are taken up by an editor, who then asks an expert to write an accessible answer to that specific question. The experts are contacted based on their scientific expertise. They are generally not trained specifically in science communication but are supplied with a one-pager with guidelines on readability.

After this writing procedure, the text is anonymously peer reviewed to increase the reliability of that answer, before being published on KlimaatHelpdesk.org. The main goal of the KH is to explain climate issues to society in a trustworthy and understandable manner, by providing popularized scientific texts. By answering questions, the KH hopes to start a dialogue between citizens and scientists.

The target audience of the KH ranges from young secondary school students to young adults (ages 13-35) with diverse

backgrounds. Therefore, the KH aims to make their answers understandable for secondary school students and up.

## 2.2 Conditions and text conversion

One text from the KH was converted into three conditions, as we aimed to separate the potential effect of the second-person voice from the first-person voice. As a basis, we chose a text on the climate impact of train versus plane travel, because it had received a big readership on the website~~,~~ (>5,000 visits), so we knew it was a popular topic, and because it was not too technical and therefore relatively easy to adapt. All three texts included the same scientific information (approximately 750 words in length and including a fairly technical figure) but differed in the number of personalization (through direct address) elements. The three texts were checked by the original author for correctness.

1.  In the first condition, the expository condition, no personalized elements were present, and the text was pallid and distant. Sentences in this text were factual and formal. For example, the text included this sentence: "A single trip from the Netherlands to Milan, about 1100 km, produces about 11 kg of $CO_2$ per person. That is less than average for train journeys in Europe, because ~~NS~~Nederlandse Spoorwegen and Deutsche Bahn (the Dutch and German national railway companies) operate mostly on wind energy, and the Swiss railways on hydropower."

2.  In the second condition, the slightly personalized condition, minor changes were made compared to the first condition. Twenty-three definite articles (e.g. "the train seat") were replaced by second-person possessive pronouns (e.g. "your train seat"). Additionally, 17 indefinite pronouns were replaced by the second-person pronoun. Such changes were done previously by Dutke *et al* [2016] and Ginns and Fraser [2010]. For example, the sentence above was changed to "With a single trip from the Netherlands to Milan, about 1100 km, you generate about 11 kg of $CO_2$ per person. That is less than average for train journeys in Europe, because ~~NS~~Nederlandse Spoorwegen and Deutsche Bahn (the Dutch and German national railway companies) operate mostly on wind energy, and the Swiss railways on hydropower."

3.  In the third condition, the highly personalized condition, the first-person voice of the writer was added. It included the same second-person (possessive) pronouns as the second condition, but also included six additional first person (plural) pronouns and thirteen direct addresses from the writer. In these direct addresses, readers were spoken to by the writer's voice. These additions made the third condition conversational instead of formal. For example, the sentence above was changed to "With a single trip from the Netherlands to Milan, about 1100 km, you generate about 11 kg of $CO_2$ per person. I think it is important to mention that this is less than the average for train journeys in Europe, because ~~NS~~Nederlandse Spoorwegen and Deutsche Bahn (the Dutch and German national railway companies) operate mostly on wind energy, and the Swiss railways on hydropower. "

The original Dutch versions of the three conditions can be found in appendix 1-3. Since Dutch is very similar to English (they share linguistic roots and numerous similarities in vocabulary, grammar, and syntax), we expect that our results are generalizable to English too.

## 2.3 Participants and study design

In the period of 20 June 2023 to 4 January 2024, we used *SurveySwap* to recruit participants. SurveySwap is an online platform [e.g., Mouratidou *et al* 2024], operating on a reciprocal basis where users can earn credits by completing other users' surveys, and then use those credits to have their own surveys completed. This system is particularly used by students and academics who need to collect a significant amount of data for their research projects or dissertations, so the pool of respondents may be limited in diversity.

Our survey was deemed low risk in the Utrecht University Ethics quick-scan, and started with consent form (based on the default template at Utrecht University, see also the ethical Statement at the end of this manuscript), in which participants were informed that their participation was voluntary and confidential, that they could stop at any moment, and that their identity and research data would not be stored together.

A total of 169 people, who all spoke Dutch, took part in our research. Participants aged younger than 18 or older than 35 were excluded from this analysis because our focus group was young adults (as these are the target audience of the KlimaatHelpdesk). Additionally, participants were excluded when the total duration time of reading the text and filling in the survey was less than 4 minutes (careless readers) or more than 30 minutes (distracted participants). This resulted in a sample size of 100 participants. We can expect (although we ~~haven't~~have not tested) that very few of the participants had previously heard of the KlimaatHelpdesk, and that even less (or none) of them had heard of the authors of the article.

Participants answered questions about their age (median=24 years; standard deviation=2.8 years), gender (44 men, 55 women, 1 other) and educational level (>50% finished higher education). Participants were randomly assigned to one of the three conditions and asked to read the text carefully and fill out a questionnaire with 9 prior and 5 posterior questions. The first condition (expository) was read by 40 participants, the second condition (slightly personalized) was read by 30 participants, and the third condition (highly personalized) was read by 30 participants.

## 2.4 Measures

### 2.4.1 Prior intention and past conduct

Prior to exposure, participants answered four questions to determine the intention and past conduct towards flying and travelling by train (Fig. 1). These questions included statements to which participants could respond on a 5-point Likert scale indicating how likely it would be that they would take the plane and train on a trip from The Netherlands to Milan (which was the topic of the KlimaatHelpdesk text used in this study). The likelihood that they would take the plane (median="very likely") was much higher than that they would take the train (median="unlikely"), with no participant answering it would be "very likely" that they would take the train to Milan. Additionally, participants answered multiple choice questions (possible answers: 0, 1, 2 and 3 or more) on how often the participants went on a vacation last year (median=2), and how many of those trips were by flying (median=1). To investigate if there was an effect of prior intention, we also separated the participants into two groups (split on the median, so that both groups were roughly equal in size): those that were "very

likely" to travel to Milan by plane (*N*=56) and those that filled out any of the other four options (*N*=44). However, since we did not find any significant effects on opinion or credibility, we do not explicitly show the results below.

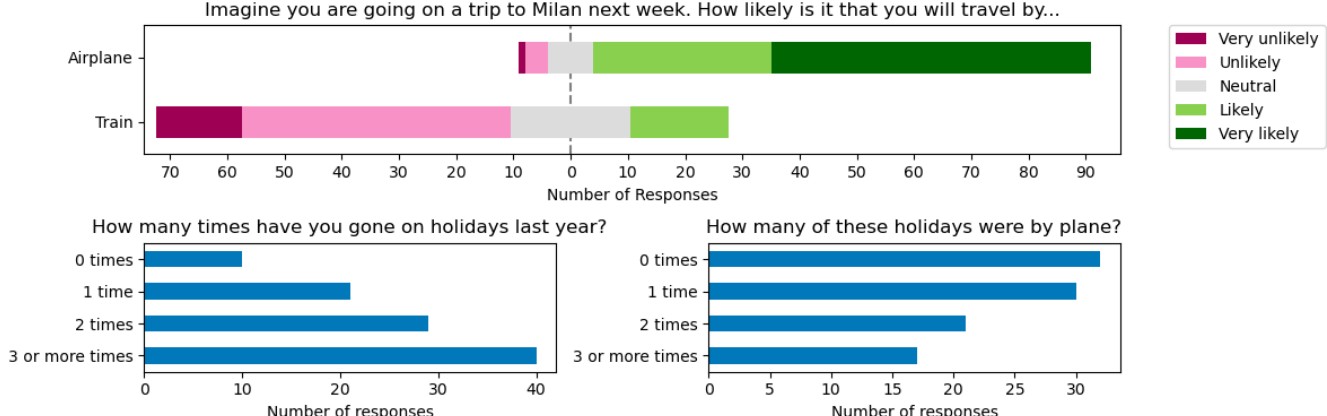

**Figure 1: Plots for prior intention and past conduct towards flying and travelling by train, as answered by the 100 participants. The upper panel uses the plot_likert python package to visualise the number of participants that have given each of the five respective answers; centered around the Neutral (Likert-score=3) value.**

### 2.4.2 Science capital and trust

The science capital of the participant was measured using four 5-point Likert scale statements (Fig. 2), retrieved from the IMPACTLAB [Peeters *et al* 2022]. The statements were "I am generally aware of new scientific discoveries and developments", "I am interested in the scientific process and the results it yields", "In my spare time, I participate in activities that allow me to learn something about science, such as visiting museums, looking up information online or watching science-related tv shows or videos", and "I regularly talk about science with other people, e.g. in my free time or in the context of my study or job".

Additionally, two 5-point Likert scale statements were added to test the prior perceived trustworthiness and intended purpose of scientists (Fig. 2). The two statements were "I generally find scientists to be trustworthy" and "I think it's important that scientists communicate about their research".

We combined the six statements into one construct "science capital and trust" (SCT). The Cronbach-Alpha score – which measures the  internal consistency [e.g., Heo *et al* 2015] of these six statements on science capital and trust – was acceptable ($\alpha$ = 0.79). Most of the participants answered "agree" or "strongly agree" on the six questions, with the largest number of (strongly) disagree answers on the "awareness" and the "talking to others" questions (Fig. 2). To investigate if there was a difference for science capital and trust on the effect and credibility, we separated the participants into two groups (split on the median, so that both groups were roughly equal in size): those that had an average score for the six science capital and trust questions of less than 4 out of 6 (*N*=53, hereafter referred to as 'SCT<4') and those that had an average score of 4 or more (*N*=47; hereafter referred to as 'SCT>=4').

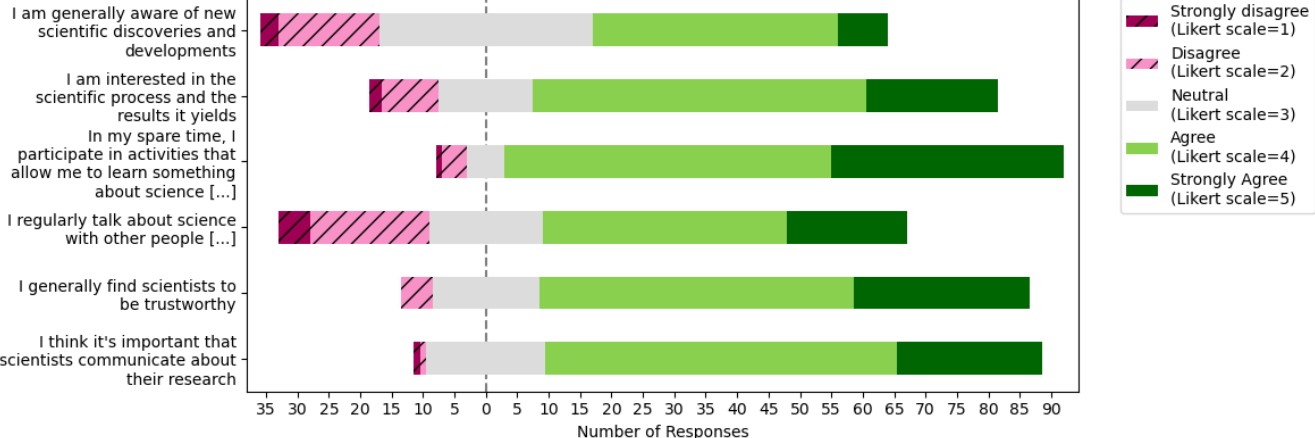

**Figure 2: The responses to the 5-point Likert statements on the science capital and trust of the participants. The participants skew highly towards high science capital and trust.**

## 2.5 Assessment of differences between conditions

To assess whether our three conditions indeed perceived to be different in personalization, we evaluated how participants experienced the text. After reading the text, participants were asked whether they found the text formal or informal and personal or professional by filling in a 10-point semantic differential scale (Fig. 3). More than 60% of the participants experienced the texts as relatively formal and professional (score <6). We separated the answers by text condition and used an ANOVA test to find that there was a significant difference in the extent to which the participants found the text personal as opposed to professional ($p$=0.003). Post hoc tests (using the Holm correction to adjust $p$ [Holm 1979]) indicated that both the slightly personalized and the highly personalized texts were perceived significantly more personal than the expository text ($p$=0.021 and $p$=0.008, respectively), but we found no evidence that the highly personalized text was perceived more personal than the slightly personalized text ($p$=0.246). There was no significant difference when we separated the responses based on science capital and trust ($p$=0.255).

There also was a significant difference in the extent to which participants found the text informal as opposed to formal (one-sided $p$=0.010). Post hoc tests (using the Holm correction to adjust $p$) indicated that both the slightly personalized and the highly personalized texts were perceived significantly more informal than the expository text (one-sided $p$=0.035 and $p$=0.030, respectively), but we found no evidence that the highly personalized text was perceived more informal than the slightly personalized text (one-sided $p$=0.908). Again, there was no significant difference when we separated the responses based on science capital and trust ($p$=0.618).

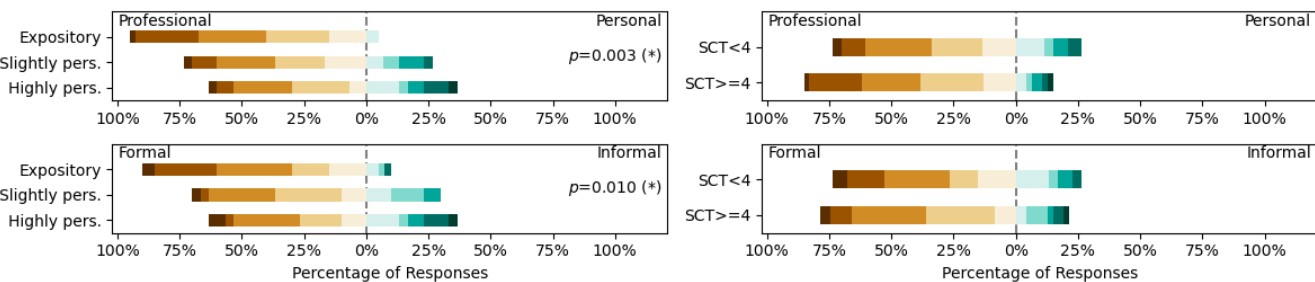

Control question: How did you experience the text?

**Figure 3: Responses to the two control questions, on a scale from 1 to 10, separated by condition (left column) and score on Science Capital and Trust (right column; higher or lower than 4 out of 5). ANOVA test statistics with p<0.05 are indicated. The color scale is such that 1-5 are brown, and 6-10 are green. As expected, the expository text was experienced as more professional and formal than the highly personalized text.**

## 3 Results

### 3.1 Change in attitude

The effect of the texts on emotions was measured using questions derived from the *IMPACTLAB* [Peeters *et al* 2022]. The first question, measuring how participants felt after reading the text, consisted of eight 10-point semantic differential statements (Fig. 4). The strongest positive response was on emotion, with >60% of the participants finding the texts interesting. There was a significant difference in the happy/unhappy emotion for all three separations, although there was no trend. Additionally, there was a significant difference for the separation based on science capital and trust in the unsatisfied/satisfied (with participants with higher SCT feeling more satisfied) and not interesting/interesting (with participants with higher SCT feeling more interested) emotions. The difference between all other emotions was not statistically significant.

Post hoc tests (using the Holm correction to adjust $p$) on the unhappy/happy emotion for the text condition indicated that the participants were significantly happier after reading both the expository and the highly personalized texts than the slightly personalized text ($p$=0.040 and $p$=0.048, respectively), but we found no evidence that participants were happier or more unhappy after reading the highly personalized text than after reading the expository text ($p$=0.662).

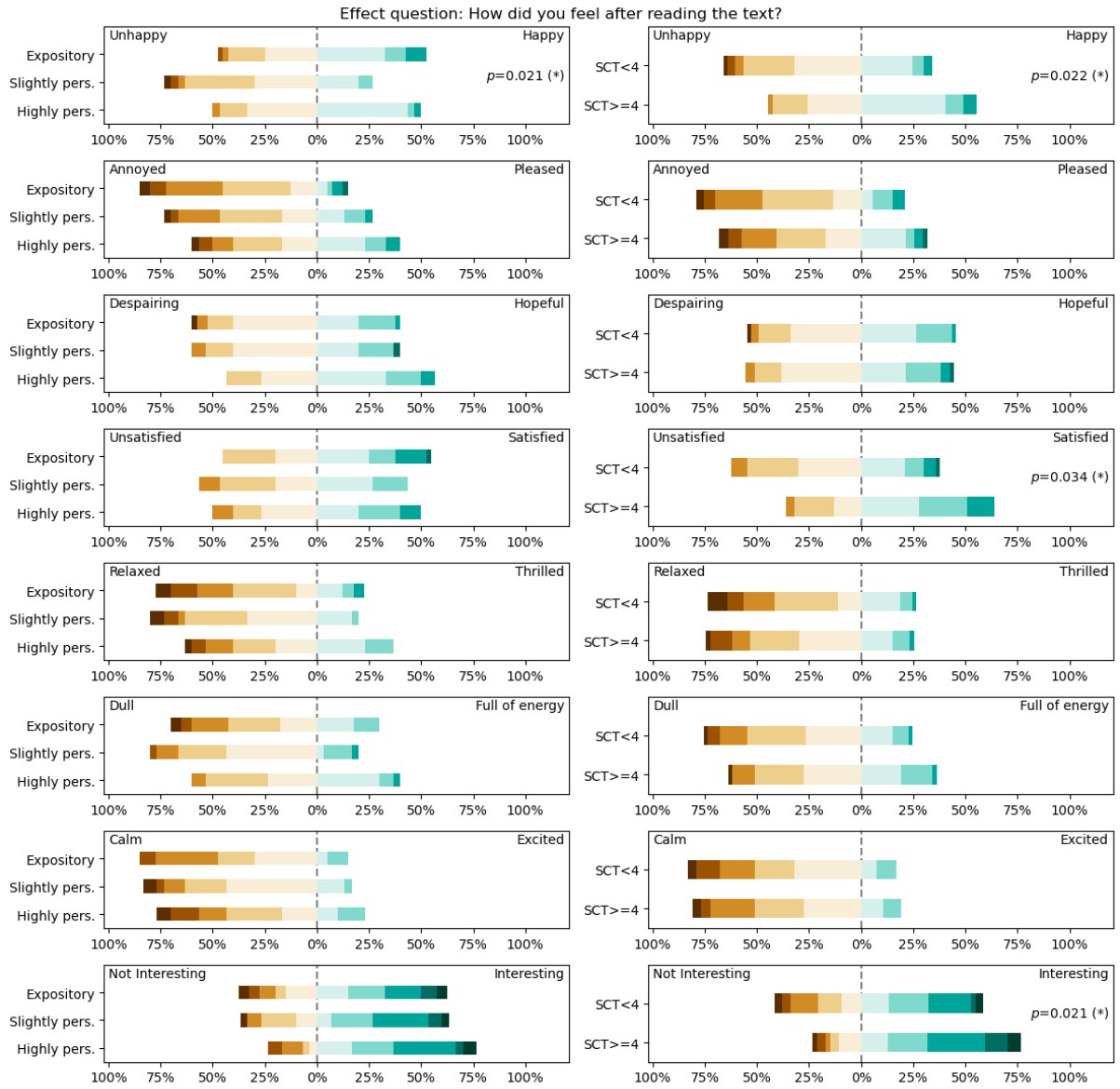

**Figure 4: Responses to the eight statements on emotions of the participants after reading the texts, on a scale from 1 to 10, separated by condition (left column), score on Science Capital and Trust (right column; higher or lower than 4 out of 5). ANOVA test statistics with p<0.05 are indicated. The color scale is such that 1-5 are brown, and 6-10 are green. There was hardly any difference between the three texts, nor between the two levels of Science Capital and Trust.**

In the second IMPACTLAB question, on the cognitive effect of the text, participants answered four statements in a 5-point Likert scale (Fig. 5): "I now know more about the impact of travel on climate"; "I want to know more about the impact of travel on climate"; "my opinion on flying or train travel has changed"; and "I want to read more of these texts - also on other scientific topics". None of these statements were answered significantly differently between the three text conditions, nor

between the two levels of science capital and trust. The question on changed opinion was also not answered statistically differently between those participants that were likely to take the plane to Milan and those that ~~weren't~~were not.

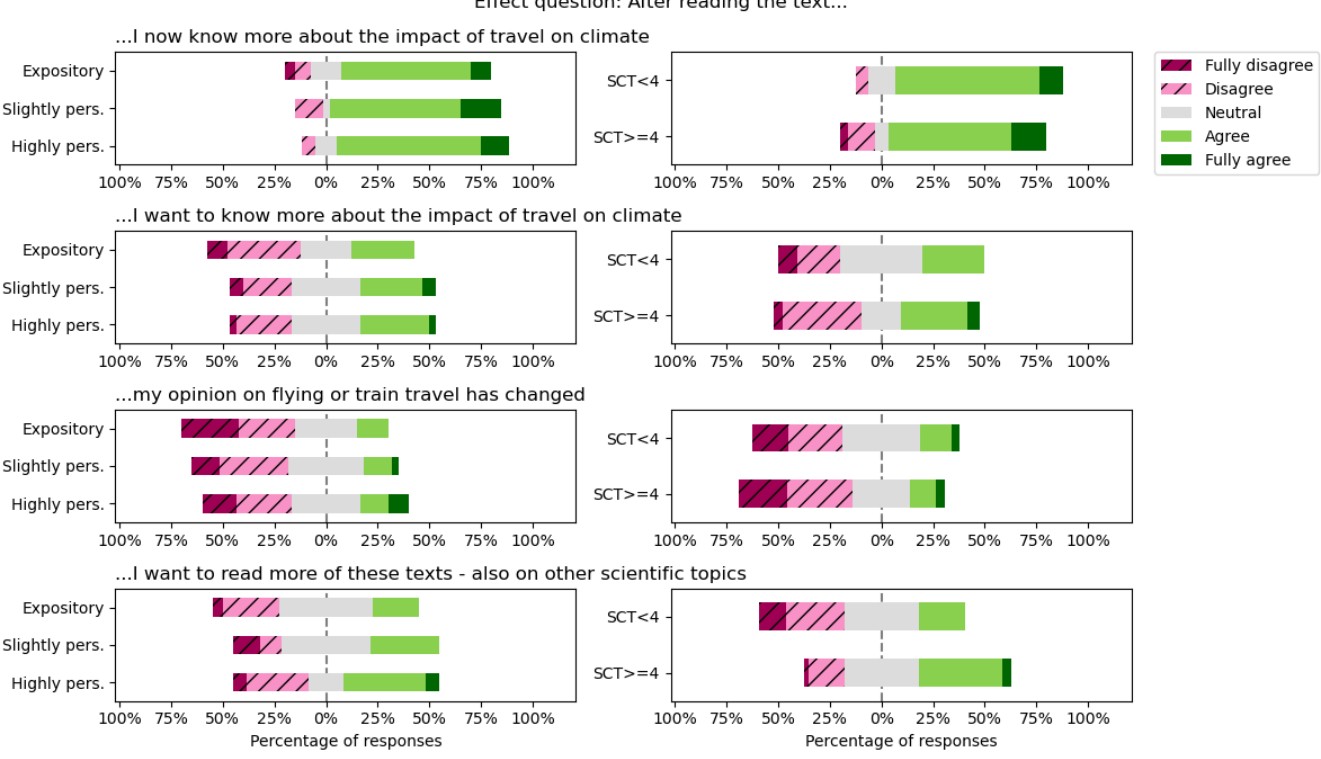

**Figure 5: Responses to the four statements on effects on the participants after reading the texts on a five-point Likert scale, separated by condition (left column), score on Science Capital and Trust (right column; higher or lower than 4 out of 5). None of the ANOVA test statistics were lower than p=0.05. There was no difference between the three texts, nor between the two levels of Science Capital and Trust.**

## 3.2 Perceived credibility of the writer

The perceived credibility of the writer was measured using eight different 7-point semantic differential statements (Fig. 6), as used by Kotcher *et al* [2017]. Randomization in the order of statements was used to prevent the results from possible order effects. Averaging these eight statements in one construct and applying an ANOVA test led to no significant difference between the three text conditions ($p$=0.502), nor a significant difference between the two levels of science capital and trust ($p$=0.116). The Cronbach-Alpha score for internal consistency of these eight statements on perceived credibility was acceptable ($\alpha = 0.73$).

Analyzing the statements individually, we again found that only two of these statements on the perceived credibility were answered significantly differently: the not at all intelligent/very intelligent question when separated by likelihood to take the

plane (p=0.011; not shown) and the not at all trustworthy/trustworthy question when separated by science capital and trust (*p*=0.019). The groups with higher SCT and lower likelihood to take the plane found the author more intelligent.

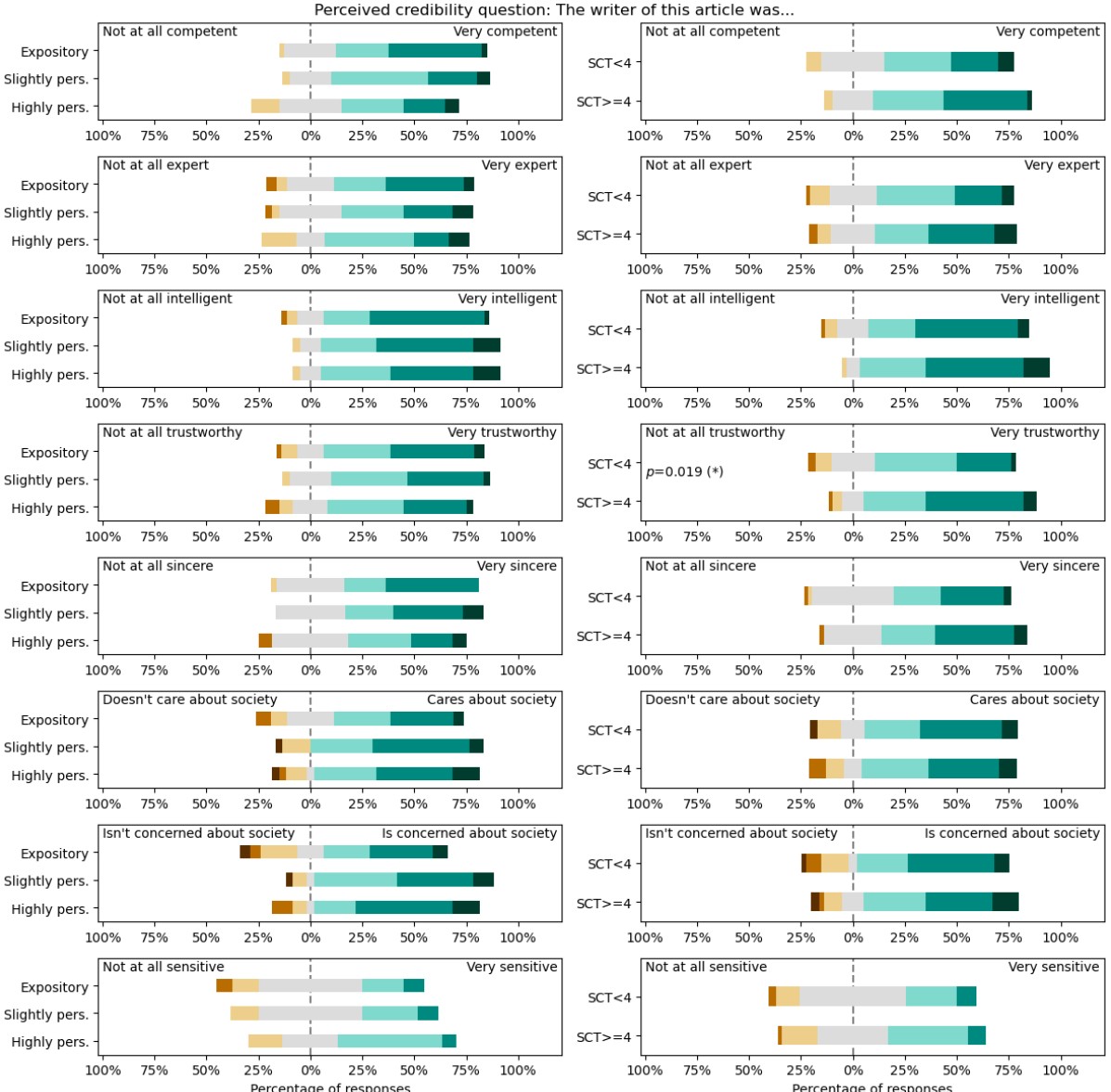

**Figure 6: Responses to the eight statements on perceived credibility of the writer, on a scale from 1 to 7, separated by condition (left column), score on Science Capital and Trust (right column; higher or lower than 4 out of 5). ANOVA test statistics with p<0.05 are indicated. The color scale is such that 1-3 are brown, 4 is grey, and 5-7 are green. Again, there was no difference between the three texts, nor between the two levels of Science Capital and Trust.**

### 3.3 Perceived credibility of the text: goal to persuade and to inform

Based on the question by Kotcher *et al* [2017], participants were then asked to what extent they agree or disagree with the following two statements: "The goal of the text was to persuade people to take action to address climate change" and "The goal of the text was to provide impartial information about travelling by airplane or train" (Fig. 7). Both statements were measured on a 7-point Likert scale (1 = Fully disagree, 7 = Fully agree), and an ANOVA test revealed that the only statistically significant result was when we separated the responses to the question on whether the writer provided impartial information by science capital and trust ($p=0.026$).

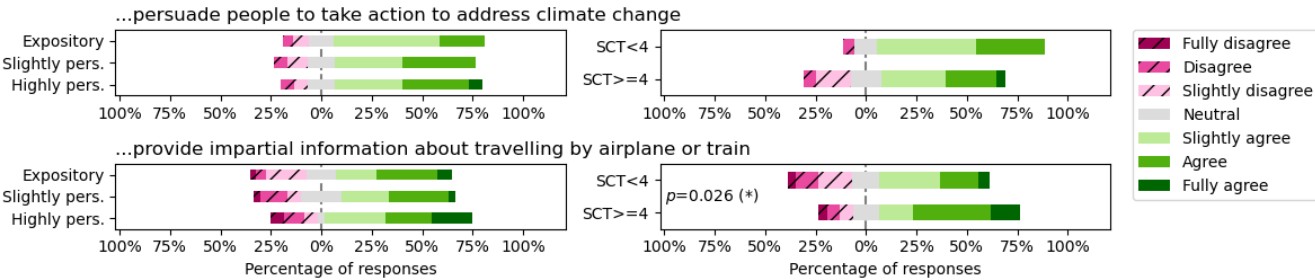

**Figure 7: Responses to the two statements on the perceived goals to persuade on a seven-point Likert scale, separated by condition (left column), score on Science Capital and Trust (right column; higher or lower than 4 out of 5). ANOVA test statistics with p<0.05 are indicated. There was no difference between the three texts, and only for impartial information between the two levels of Science Capital and Trust.**

### 3.4 Perceived credibility of the writer: attribution to scientific evidence and political views

Also based on the questions by Kotcher *et al* [2017], participants were finally asked to what extent they agree or disagree with the following two statements: "The content of the text was shaped by the writer's evaluation of the scientific evidence about the impact of travelling by airplane or train on the environment" and "The content of the text was shaped by the writer's personal views about the impact of travelling by airplane or train on the environment" (Fig. 8). Both statements were measured on a 7-point Likert scale (1 = Fully disagree, 7 = Fully agree). The first statement about scientific evidence was answered statistically differently between the science capital and trust groups ($p=0.017$) and the second statement about personal views was answered statistically differently between the three text conditions ($p=0.041$).

Post hoc tests (using the Holm correction to adjust $p$), however, revealed no evidence that participants found that the content was shaped by the writer's personal views after reading the expository text compared to the slightly personalized text ($p=0.880$), after reading the expository text compared to the highly personalized text ($p=0.071$), or after reading the slightly personalized text compared to the highly personalized text ($p=0.071$).

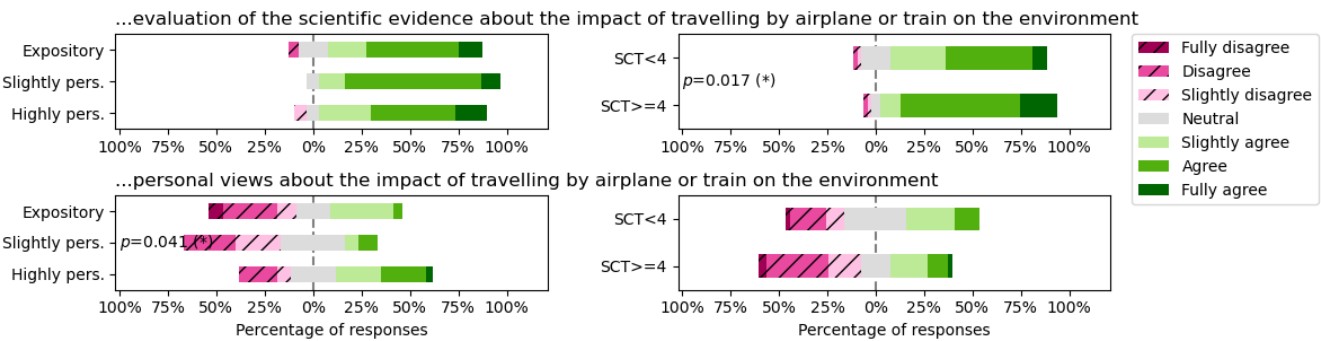

**Figure 8: Responses to the two statements on the perceived goals to persuade on a seven-point Likert scale, separated by condition (left column), score on Science Capital and Trust (right column; higher or lower than 4 out of 5). ANOVA test statistics with p<0.05 are indicated. There was hardly any between the three texts, nor between the two levels of Science Capital and Trust.**

## 4 Discussion

In this study, we set out to gain insights into the effects of personalization in writing about traveling by train or plane on the perceived credibility of the writer (the scientist). The variation in the amount of personalization in the texts was recognized by readers (using second-person pronouns in the 'slightly personalised' condition; and adding the first-person voice of the author in the 'highly personalized' condition), as was apparent from the result that the highly personalized version of the text was perceived much more personal and informal than the base (expository) version (Fig. 3). In that sense, our textual changes worked as intended; although it should be noted that most participants (>60%) experienced even the highly personalized text as relatively formal and professional (score <6). We thus did not manage to rewrite the expository text to such an extent to move it from majority-formal to majority-informal while keeping the content the same. It may be that our manipulation, in our attempt to keep the text as similar as possible, has been too subtle and that, in real life, when scientists write more personalized texts, changes in the tone and content affects the text more than we have operationalized in this study. Additionally, readers may generally be likely to perceive a text as formal if it contains scientific information.

The answer to our first Research Question ("*How does personalization of popular scientific climate texts affect the interest and opinion to climate change of participants?*") is that we see a limited effect (Fig. 5). Most participants indicated they know more about the impact of travel on climate after reading the text, independent of the text version they read, and independent of their science capital. On the other hand, reading the texts did not change most of the participant's opinions on flying or train travel, again with no difference between texts nor science capital. It could be that some of the participant's attitudes to e.g. the need for a better rail infrastructure has changed; but since we only asked about their opinion on flying or train travel, we ~~can't~~cannot evaluate that effect.

Of course, in real life, people will be exposed to various sources of information about this issue and previous research has shown that especially peer pressure can be effective in changing behaviors. Since KH is essentially an interactive question-

answer based website, future research may investigate if adding simulated responses from other readers (indicating that, based on what they just learned, they would take the train) would have more effect. This concept of communicated actions of peers (even anonymously) having a positive effect on behavior has been shown in various contexts already, among which preventive health behavior [Saran *et al* 2018].

The questions on the effects of the texts on emotions (Fig. 4) did not vary significantly when we separated by text condition, nor by science capital and trust, except for the emotion of happiness which was significant for both types of separation. This confirms the findings by Peeters *et al* [2022] that happiness is one of the strongest predicting emotions for effect. Most participants found the text interesting (>60% for all three conditions), but felt calm, dull, and relaxed (i.e. not excited) after reading. Additionally, they also felt annoyed. We conclude that the participants did not enjoy reading any of the three conditions and this may also indicate that a more extreme shift between conditions is necessary to better simulate personalized science texts aiming to entice.

As for the answer to our second Research Question ("*What is the effect of personalization on the perceived credibility of a popular scientific text and the climate scientist that wrote it?*"): the writer of the article was perceived very positively (Fig. 6) as competent, expert, very intelligent, very trustworthy etc. The only statement where the writer did not score more than 60% positive was on the element of sensitivity, although most responses there were close to neutral. This seems to indicate that participants' attitude toward the writer was generally positive, even though most participants did perceive that the goal of the text was to persuade people to take climate action. The positive attitude towards the writer (Figure 2) might reflect the general trust in scientists as a source of information [Edelman Trust Institute 2024]. Participants with high Science Capital and Trust more strongly perceived the goal of the writer to provide impartial information; compared to participants with lower SCT. Perhaps surprisingly, participants did not perceive the text to be more shaped by the writer's personal views in the highly personalized condition (Fig. 8), despite it being perceived more personal (Fig. 3). This may indicate a weak manipulation, although we would argue that instead, this might suggest that a more personal text does not influence the credibility of the writer. This is also in line with past research that shows that this mode of communication, in a more activistic tone, does not necessarily hamper public perceptions of scientific integrity and scientists' credibility [Kotcher *et al* 2017, Cologna *et al* 2021].

Previous research has shown that scientists tend to stick to the facts and create messages with too much detail and a lack of personal connection [Somerville and Hassol 2011]. Therefore, an important recommendation is to make personally-relevant messages by communicating on the level of values [Seethaler *et al* 2019, Clarke *et al* 2020, Fage-Butler *et al* 2022]. Given the fact that our base text was purposefully expository, it may be that the operationalization of these values was quite weak and that adding a more explicit incorporation of the writer's personal values, would have had a bigger effect on the readers.

Of course, this study has limitations. First, the sample size and characteristics of the used conditions limit our options to generalize outside of this group. Especially our sample size restricts our analysis making it impossible to show small differences across groups. A larger sample size might make some of the more subtle, non-significant details in the Figures more pronounced. The benefit is that the differences reported are robust, the downside is that there may be hidden effects.

Also, most participants were relatively highly educated, which may also have influenced a greater acceptance of the expository condition as when the science capital of the participants would have been more diversified. Furthermore, the required time to finish the survey was quite long (8 minutes on average). This may have caused participants to be less attentive and engaged, especially toward the end of the questionnaire; and could thus explain that the last few questions showed fewer significant difference between the conditions. Finally, we did not make changes to the visual cues in the article. All three conditions contained a figure that was fairly technical, which can be expected to minimize the personalization effects of the text, giving the overall look of the article a more expository feel. Future research could investigate this further by altering the visuals in accordance with the text-based conditions, although a 3 (visuals) by 3 (text) design would necessitate even more respondents.

## 5 Conclusion

In this study, we analyzed how variation in personalization through direct address in an article about the effect of travelling by train or plane on carbon dioxide emissions affected opinion and interest toward sustainable travel and the perceived credibility of the climate scientist who wrote it specifically. We used an article that was previously published on the KlimaatHelpdesk.org platform and adapted that to increase personalization. To measure the effect, we used a questionnaire with questions that were previously validated by Kotcher *et al* [2017] and Peeters *et al* [2022]. Our findings show that a limited amount of personalization of the text was recognized and positively appreciated by the readers and did not affect the credibility of the writer.

Of course, this is only one study on one text with one type of audience (Dutch young adults). If our results hold up in a wider variety of texts and audiences, this suggests that adding personalization does not harm the message in climate communication materials, which is a useful finding for communication professionals who aim to make climate texts more engaging.

## Acknowledgements

Funding for this project was provided through an Agnites Vrolik Award by the Utrecht University Fund. We thank Aike Vonk, Tugce Varol and Nieske Vergunst for insightful comments on a draft of this manuscript and Frances Wijnen for co-supervision of the initial stage of the project.

## Code/Data availability

The stacked bar graph plots were made using the plot_likert library, distributed under a BSD-3 license at https://github.com/nmalkin/plot-likert/. In the spirit of Open Science, all data and scripts used for the manuscript are available at https://doi.org/10.5281/zenodo.12579018.

## Author contributions

AL designed the survey and wrote the first draft of the manuscript. EvS analyzed the data from the survey and wrote the second draft of the manuscript. All authors designed the study and edited the manuscript.

## Competing interests

DR is an editor at the KlimaatHelpdesk and EvS is an ambassador for the KlimaatHelpdesk.

## Ethical statement

Conform Utrecht Univertity's Science-Geo Ethics Review Board protocol, an ethics and privacy QuickScan was conducted to verify if ethical considerations had to be taken during the study. Based on the study design, this study was classified low-risk and therefore no further ethical review or privacy assessment was required.

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

## Hoe milieuonvriendelijk is vliegen tegenover de trein als je de aanleg en onderhoud van infrastructuur meerekent?

De trein wint in alle opzichten van het vliegtuig. De directe $CO_2$-uitstoot, dus tijdens het vervoer, is gemiddeld lager, en ook als de bouw van de infrastructuur wordt meegerekend wint de trein. En omdat het vliegtuig veel sneller is verleidt het om veel verdere reizen te maken, wat ook tot meer uitstoot leidt. Van een omslagpunt in afstand, zoals sommigen suggereren, is geen sprake: hoe verder de vlucht, hoe groter de $CO_2$-uitstoot. De treinreiziger blijft daar altijd substantieel onder.

### Duurzaamheid is meer dan alleen $CO_2$-emissie

De milieubelasting van reizen wordt berekend door de gereisde afstand, het aantal reizen en de $CO_2$-emissiefactor met elkaar te vermenigvuldigen. De emissiefactor zegt zelf weinig over de totale emissies, laat staan de 'duurzaamheid'. Een enkele treinreis vanuit Nederland naar Milaan, zo'n 1100 km, veroorzaakt ongeveer 11 kg $CO_2$ per persoon. Dat is minder dan gemiddeld voor treinreizen in Europa, omdat de NS helemaal en Deutsche Bahn grotendeels op windenergie rijden en de Zwitsers op waterkracht. Alleen in Italië zorgt fossiele stroom voor hogere emissies. De gemiddelde treinreis van 1100 km komt op zo'n 25 kg $CO_2$. Voor een vliegreis naar Milaan is de uitstoot 87 kg voor een enkele reis (8x meer). Een even lange reistijd met het vliegtuig als de treinreis naar Milaan duurt, bijvoorbeeld een vlucht naar Mumbai, zorgt voor 560 kg $CO_2$ uitstoot (50x meer). Dit alles op basis van de $CO_2$ die ontstaat tijdens de reis. Reisgedrag, dus hoe vaak én hoever iemand reist, is dan ook ontzettend belangrijk voor duurzaamheid.

### Hoe zit dat met de emissies voor infrastructuur?

De resultaten van onderzoek variëren nogal. Zo concluderen Chester en Horvath dat de $CO_2$-emissies voor bouw en onderhoud van infrastructuur en voor productie van brandstof de gemiddelde emissiefactor van het vliegtuig met 31% verhogen en die van de trein met 155% [2]. Die ophoogfactoren lijken wel erg hoog, zeker voor de trein. In het voorbeeld naar Milaan komt dat neer op 29 kg voor de treinreis en 94 kg voor de vliegreis. Dat komt mogelijk omdat deze cijfers uitgaan van nog weinig intensief

gebruikte light-rail systemen. Op basis van de cijfers van een hogesnelheidslijn in Zweden blijkt de uitstoot met infrastructuur bij 1 miljoen reizigers per jaar ongeveer 0,07 kg $CO_2$ per passagierskilometer (kg/pkm) te zijn, waarmee de treinreis naar Milaan ongeveer gelijk zou uitkomen met de vliegreis (0,07 kg/pkm * 1100 km = 11 kg voor het rijden van de trein = 88 kg $CO_2$, vergelijk met de 87 kg van hierboven zonder infrastructuur) [1]. Bij 10 miljoen reizigers is de emissiefactor voor infrastructuur en onderhoud nog maar 0,009 kg/pkm (één treinreis naar Milaan komt dan op 0,009 kg/pkm * 1100 km +11 km = 20,9 kg $CO_2$); veel minder dan voor het vliegtuig zonder infrastructuur, laat staan mét. Gemiddeld voor de Europese hogesnelheidstreinen is het zelfs maar 0,006 kg/pkm (6,6 kg $CO_2$) [5].

### Het effect van afstand en niet meereizen

Terug naar het effect van afstand op de directe $CO_2$-emissies. De twee grafieken hieronder laten zien dat de emissies per passagierskilometer van het vliegtuig weliswaar lager worden met langere afstanden (linker figuur), maar de totale emissies van de reis gewoon toenemen met de afstand (rechter figuur). Ook voor de trein geldt natuurlijk dat verder reizen tot meer uitstoot leidt. En dat is het enige dat telt voor het klimaat.

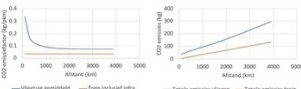

Grafieken op basis van bewerkte data uit Eijgelaar et al. 2021.

Aangezien infrastructuuremissies in het verleden zijn ontstaan is het bij het kiezen tussen de trein en het vliegtuig overigens ook belangrijk naar de 'marginale emissies' te kijken. Daarmee wordt bedoeld hoeveel emissies bespaard worden door niet met het vliegtuig of de trein te reizen. Één treinreiziger minder maakt de trein iets lichter en gebruikt misschien minder stroom omdat er geen oplader gebruikt wordt. De op hele reis naar Milaan komt dat misschien op een half kilogrammetje $CO_2$-uitstoot neer. Bij het vliegtuig zijn de marginale emissies groter, want 'niet-meegaan' bespaart juist wél brandstof en dus emissies. Dat komt omdat een vliegtuig veel gevoeliger is voor extra gewicht omdat de vleugel dan meer draagkracht moet leveren. Één passagier minder op een vlucht van 10.000 km in een moderne Boeing 787 Dreamliner bespaart 90 kg $CO_2$ directe uitstoot! Voor een vlucht naar Milaan is dat nog altijd 15 kg $CO_2$ per lege stoel,

dertig keer zoveel als de extra emissies door een extra bezette stoel in de trein naar Milaan.

### Hoe kwam dit artikel tot stand?

Dit antwoord is geschreven door Paul Peeters
Reviewer: José Potting
Redacteur: Joseline Houwman
Gepubliceerd op: 21-9-2021
Wat vond je van dit antwoord? Geef ons je mening!

### Bronnen

[1] Åkerman, J. (2011). The role of high-speed rail in mitigating climate change - The Swedish case Europabanan from a life cycle perspective. Transportation Research Part D: Transport and Environment, 16(3), 208-217.

[2] Chester, M. V., & Horvath, A. (2009). Environmental assessment of passenger transportation should include infrastructure and supply chains Environmental Research Letters, 4(024008), 1-8.

[3] Kuosmanen, T., & Kuosmanen, N. (2009). How not to measure sustainable value (and how one might). Ecological Economics, 69(2), 235-243.

[4] Tillman, A.-M., Ekvall, T., Baumann, H., & Rydberg, T. (1994). Choice of system boundaries in life cycle assessment. Journal of Cleaner Production, 2(1), 21-29.

[5] Tuchschmid, M. (2009). Carbon Footprint of High-Speed railway infrastructure (Pre-Study). Methodology and application of High Speed railway operation of European Railways. Zürich: The International Union of Railways (UIC).

[6] Eijgelaar, E., Peeters, P. M., Neelis, I., De Bruijn, K., & Dirven, R. (2021). Travelling Large in 2019. The Carbon Footprint of Dutch Holidaymakers in 2019 and the Development since 2002 (ISBN: 978-90-825477-6-4).

**Figure A1: the expository condition. This image was created by the authors, adapted with approval from the original version at https://www.klimaathelpdesk.org/answers/hoe-milieuonvriendelijk-is-vliegen-tegenover-trein-of-bus-met-aanleg-en-onderhoud-van-infrastructuur-meegerekend/, which was published with a CC BY-NC-SA 4.0 license.**

440

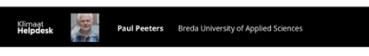

**Paul Peeters** — Breda University of Applied Sciences

# Hoe milieuvriendelijk is vliegen tegenover trein als je de aanleg en onderhoud van infrastructuur meerekent?

De trein wint in alle opzichten van het vliegtuig. Je directe $CO_2$-uitstoot, dus tijdens het vervoer, is gemiddeld lager, en ook als je de bouw van de infrastructuur meerekent wint de trein. En omdat het vliegtuig veel sneller gaat, word je snel verleid om veel verdere reizen te maken, wat ook tot meer uitstoot leidt. Van een omslagpunt in afstand, zoals sommigen suggereren, is geen sprake: hoe verder je vlucht, hoe groter je $CO_2$-uitstoot. Als je met de trein reist blijf je daar altijd substantieel onder.

## Duurzaamheid is meer dan alleen $CO_2$-emissie

De milieubelasting van je reis kan je berekenen door je gereisde afstand, het aantal reizen en de $CO_2$-emissiefactor met elkaar te vermenigvuldigen. De emissiefactor zegt zelf weinig over de totale emissies tijdens jouw reis, laat staan de 'duurzaamheid'. Met een enkele treinreis vanuit Nederland naar Milaan, zo'n 1100 km, veroorzaak je ongeveer 11 kg $CO_2$. Dat is minder dan gemiddeld voor treinreizen in Europa, omdat de NS helemaal en Deutsche Bahn grotendeels op windenergie rijden en de Zwitsers op waterkracht. Alleen in Italië zorgt fossiele stroom voor hogere emissies. Wanneer je naar Milaan vliegt is je uitstoot 87 kg voor een enkele reis (8x meer). Een even lange reistijd met het vliegtuig als de treinreis naar Milaan duurt, bijvoorbeeld als je naar Mumbai vliegt, zorgt voor 560 kg $CO_2$ uitstoot (50x meer). Dit alles op basis van de $CO_2$ die ontstaat tijdens je reis. Jouw reisgedrag, dus hoe vaak én hoever je reist, is dan ook ontzettend belangrijk voor duurzaamheid.

## Hoe zit dat met de emissies voor infrastructuur?

De resultaten van onderzoek variëren nogal. Zo concluderen Chester and Horvath dat de $CO_2$-emissies voor bouw en onderhoud van infrastructuur en voor productie van brandstof, de gemiddelde emissiefactor van het vliegtuig met 31% verhogen en die van de trein met 155% [2]. Die ophoogfactoren lijken wel erg hoog, zeker voor de trein. In het voorbeeld naar Milaan kom je dan op 29 kg voor je treinreis en 94 kg voor je vliegreis. Dat komt mogelijk omdat deze cijfers uitgaan van nog weinig intensief gebruikte light-rail systemen. Op basis van de cijfers van een hogesnelheidslijn in Zweden blijkt de uitstoot met infrastructuur bij 1 miljoen reizigers per jaar ongeveer 0,07 kg $CO_2$ per passagierskilometer (kg/pkm) te zijn. Daarmee komt je treinreis naar Milaan ongeveer gelijk uit met je vliegreis (0,07 kg/pkm * 1100 km + 11 kg voor het rijden van de trein = 88 kg $CO_2$; vergelijk met de 87 kg van hierboven zonder infrastructuur) [1]. Wanneer je één van 10 miljoen reizigers bent, is de emissiefactor voor infrastructuur en onderhoud nog maar 0,009 kg/pkm (je treinreis naar Milaan komt dan op 0,009 kg/pkm * 1100 km + 11 kg = 20,9 kg $CO_2$; veel minder dan je uitstoot met het vliegtuig zonder infrastructuur, laat staan mét. Gemiddeld voor de Europese hogesnelheidstreinen berekende Tuchschmid dat je uitstoot zelfs maar 0,006 kg/pkm is (6,6 kg $CO_2$) [5].

## Het effect van afstand en niet meereizen

Terug naar het effect van afstand op je directe $CO_2$-emissies. De twee grafieken hieronder laten je zien dat de emissies per passagierskilometer van het vliegtuig weliswaar lager worden met langere afstanden (linker figuur), maar dat de totale emissies van je reis gewoon toenemen met de afstand (rechter figuur). Ook voor de trein geldt natuurlijk dat je, wanneer je verder reist, meer uitstoot. En dat is het enige dat telt voor het klimaat.

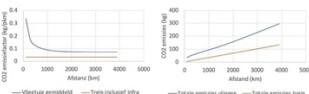

*Grafieken op basis van bewerkte data uit Eijgelaar et al. 2021.*

Aangezien infrastructuuremissies in het verleden zijn ontstaan is het bij je keuze tussen de trein en het vliegtuig overigens ook belangrijk dat je naar de 'marginale emissies' kijkt. Daarmee wordt bedoeld hoeveel emissies je kan besparen door niet met het vliegtuig of de trein te reizen. Eén treinreiziger minder maakt de trein iets lichter en gebruikt misschien minder stroom omdat je geen oplader gebruikt wordt. Bij het vliegtuig zijn de marginale emissies groter, want met 'niet-meegaan' bespaar je juist wel brandstof en dus emissies. Dat komt omdat een vliegtuig veel gevoeliger is voor je extra gewicht, omdat de vleugel dan meer draagkracht moet leveren. Eén passagier minder op een vlucht van 10.000 km in een moderne Boeing B787 Dreamliner bespaart 90 kg $CO_2$ directe uitstoot! Voor je vlucht naar Milaan is dat nog altijd 15 kg $CO_2$ voor je lege stoel, dertig keer zoveel als de extra emissies door je extra bezette stoel in de trein naar Milaan.

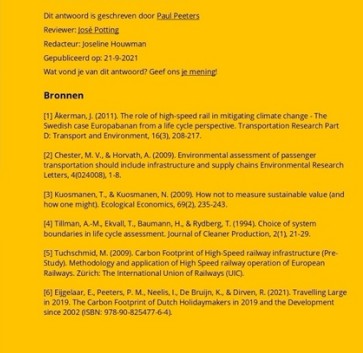

## Hoe kwam dit artikel tot stand?

Dit antwoord is geschreven door Paul Peeters
Reviewer: José Potting
Redacteur: Joseline Houwman
Gepubliceerd op: 21-9-2021
Wat vond je van dit antwoord? Geef ons je mening!

### Bronnen

[1] Åkerman, J. (2011). The role of high-speed rail in mitigating climate change - The Swedish case Europabanan from a life cycle perspective. Transportation Research Part D: Transport and Environment, 16(3), 208-217.

[2] Chester, M. V., & Horvath, A. (2009). Environmental assessment of passenger transportation should include infrastructure and supply chains Environmental Research Letters, 4(024008), 1-8.

[3] Kuosmanen, T., & Kuosmanen, N. (2009). How not to measure sustainable value (and how one might). Ecological Economics, 69(2), 235-243.

[4] Tillman, A.-M., Ekvall, T., Baumann, H., & Rydberg, T. (1994). Choice of system boundaries in life cycle assessment. Journal of Cleaner Production, 2(1), 21-29.

[5] Tuchschmid, M. (2009). Carbon Footprint of High-Speed railway infrastructure (Pre-Study). Methodology and application of High Speed railway operation of European Railways. Zürich: The International Union of Railways (UIC).

[6] Eijgelaar, E., Peeters, P. M., Neelis, I., De Bruijn, K., & Dirven, R. (2021). Travelling Large in 2019. The Carbon Footprint of Dutch Holidaymakers in 2019 and the Development since 2002 (ISBN: 978-90-825477-6-4).

**Figure A2: the slightly personalised condition. This image was created by the authors, adapted with approval from the original version at https://www.klimaathelpdesk.org/answers/hoe-milieuonvriendelijk-is-vliegen-tegenover-trein-of-bus-met-aanleg-en-onderhoud-van-infrastructuur-meegerekend/, which was published with a CC BY-NC-SA 4.0 license.**

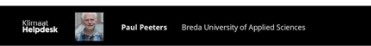

Klimaat Helpdesk — **Paul Peeters** — Breda University of Applied Sciences

# Hoe milieuonvriendelijk is vliegen tegenover trein als je de aanleg en onderhoud van infrastructuur meerekent?

De trein wint het in alle opzichten van het vliegtuig en ik ga je hier uitleggen waarom. Ik heb uitgerekend dat je directe $CO_2$-uitstoot, dus tijdens het vervoer, gemiddeld lager is. En ook als we de bouw van de infrastructuur meerekenen wint de trein. En omdat het vliegtuig veel sneller is verleidt het ons om veel verdere reizen te maken, wat ook tot meer uitstoot leidt. Van een omslagpunt in afstand, zoals sommigen suggereren, is geen sprake: hoe verder je vlucht, hoe groter je $CO_2$-uitstoot. Met de trein reizen is altijd beter voor het milieu.

## Duurzaamheid is meer dan alleen $CO_2$-emissie

Laat me je vertellen hoe je de milieubelasting van je reizen kan berekenen, namelijk door de gereisde afstand, het aantal reizen en de $CO_2$-emissiefactor met elkaar te vermenigvuldigen. De emissiefactor zegt zelf weinig over de totale emissies tijdens jouw reis, laat staan de 'duurzaamheid'. Ik heb de berekeningen hier voor je toegepast. Met een enkele treinreis vanuit Nederland naar Milaan, zo'n 1100 km, veroorzaak je ongeveer 11 kg $CO_2$. Ik vind het belangrijk te benoemen dat dat minder is dan gemiddeld voor treinreizen in Europa, omdat de NS helemaal en Deutsche Bahn grotendeels op windenergie rijden en de Zwitsers op waterkracht. Alleen in Italië zorgt fossiele stroom voor hogere emissies. Wanneer je naar Milaan vliegt is je uitstoot 87 kg voor een enkele reis (8x meer). Maar, tijd speelt ook een belangrijke rol. Ik heb voor je berekend wat de reistijd betekent voor de uitstoot. Een even lange reistijd met het vliegtuig als de treinreis naar Milaan, bijvoorbeeld als je naar Mumbai vliegt, zorgt voor 560 kg $CO_2$ uitstoot (50x meer). Dat is echt een behoorlijk grote uitstoot. Dit alles is op basis van de $CO_2$ die ontstaat tijdens je reis zelf. Ons reisgedrag, hoe vaak én hoever we reizen, is dan ook ontzettend belangrijk voor duurzaamheid.

## Hoe zit dat met de emissies voor infrastructuur?

Tijdens mijn zoektocht naar de emissies voor infrastructuur, zag ik dat de resultaten van eerder onderzoek nogal variëren. Zo concluderen mijn collega's Chester and Horvath dat de $CO_2$-emissies voor bouw en onderhoud van infrastructuur en voor productie van brandstof de gemiddelde emissiefactor van het vliegtuig met 31% verhogen en die van de trein met 155% [2]. Die ophoogfactoren lijken me wel erg hoog, zeker voor de trein. In het voorbeeld naar Milaan kom je dan op 29 kg voor je treinreis en 94 kg voor je vliegreis. Dat komt mogelijk omdat deze cijfers uitgaan van nog weinig intensief gebruikte light-rail systemen. Zo vind ik dat, op basis van de cijfers van een hogesnelheidslijn in Zweden, de uitstoot met infrastructuur bij 1 miljoen reizigers per jaar ongeveer 0,07 kg $CO_2$ per passagierskilometer (kg/pkm) is. Daarmee komt je treinreis naar Milaan ongeveer gelijk uit met je vliegreis (0,07 kg/pkm * 1100 km + 11 kg voor het rijden van de trein = 88 kg $CO_2$; vergelijk met de 87 kg van hierboven zonder infrastructuur) [3]. Wanneer je één van 10 miljoen reizigers bent, is de emissiefactor voor infrastructuur en onderhoud nog maar 0,009 kg/pkm (je treinreis naar Milaan komt dan op 0,0009 kg/pkm * 1100 km + 11 kg = 20,9 kg $CO_2$); veel minder dan je uitstoot met het vliegtuig zonder infrastructuur, laat staan mét. Gemiddeld voor de Europese hogesnelheidstreinen berekende Tuchschmid dat je uitstoot zelfs maar 0,006 kg/pkm is (6,6 kg $CO_2$) [5]. Als we de aarde dus willen beschermen, dan kunnen we, op basis van deze gegevens, het beste met de trein reizen.

## Het effect van afstand en niet meereizen

Laat ik teruggaan naar het effect van afstand op je directe $CO_2$-emissies. De twee grafieken hieronder laten je zien dat de emissies per passagierskilometer van het vliegtuig weliswaar lager worden met langere afstanden (linker figuur), maar de totale emissies van je reis gewoon toenemen met de afstand (rechter figuur). Ook voor de trein geldt natuurlijk dat je, wanneer je verder reist, meer uitstoot. Helaas is dat het enige dat telt voor het klimaat. Ik kan het niet mooier maken dan het is.

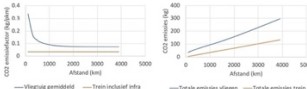

Grafieken op basis van bewerkte data uit Eijgelaar et al. 2021.

Aangezien infrastructuuremissies al in het verleden zijn ontstaan, is het bij het kiezen tussen de trein en het vliegtuig overigens ook belangrijk dat je naar de 'marginale emissies' kijkt. Daarmee wordt bedoeld hoeveel emissies je kan besparen door niet met het vliegtuig of trein te reizen. Één treinreiziger minder maakt de trein iets lichter en gebruikt misschien minder stroom, omdat er geen oplader gebruikt wordt, wat op een hele reis naar Milaan misschien een half kilogrammetje $CO_2$-uitstoot neerkomt. Bij het vliegtuig zijn de marginale emissies groter, want met 'niet-meegaan' bespaar je juist veel brandstof en dus emissies. Dat komt omdat een vliegtuig veel gevoeliger is voor extra gewicht, omdat de vleugel dan meer draagkracht moet leveren. Één passagier minder op een vlucht van 10.000 km in een moderne Boeing B787 Dreamliner bespaart 90 kg $CO_2$ directe uitstoot. Voor je vlucht naar Milaan is dat nog altijd 15 kg $CO_2$ voor de lege stoel, dertig keer zoveel als de extra emissies door de lege stoel in de trein naar Milaan. Dus, als je voor het milieu kiest, kies je voor de trein.

### Hoe kwam dit artikel tot stand?

Dit antwoord is geschreven door Paul Peeters
Reviewer: José Potting
Redacteur: Joseline Houwman
Gepubliceerd op: 21-9-2021
Wat vond je van dit antwoord? Geef ons je mening!

### Bronnen

[1] Åkerman, J. (2011). The role of high-speed rail in mitigating climate change - The Swedish case Europabanan from a life cycle perspective. Transportation Research Part D: Transport and Environment, 16(3), 208-217.

[2] Chester, M. V., & Horvath, A. (2009). Environmental assessment of passenger transportation should include infrastructure and supply chains Environmental Research Letters, 4(024008), 1-8.

[3] Kuosmanen, T., & Kuosmanen, N. (2009). How not to measure sustainable value (and how one might). Ecological Economics, 69(2), 235-243.

[4] Tillman, A.-M., Ekvall, T., Baumann, H., & Rydberg, T. (1994). Choice of system boundaries in life cycle assessment. Journal of Cleaner Production, 2(1), 21-29.

[5] Tuchschmid, M. (2009). Carbon Footprint of High-Speed railway infrastructure (Pre-Study). Methodology and application of High Speed railway operation of European Railways. Zürich: The International Union of Railways (UIC).

[6] Eijgelaar, E., Peeters, P. M., Neelis, I., De Bruijn, K., & Dirven, R. (2021). Travelling Large in 2019. The Carbon Footprint of Dutch Holidaymakers in 2019 and the Development since 2002 (ISBN: 978-90-825477-6-4).

445

**Figure A3: the highly personalized condition.** This image was created by the authors, adapted with approval from the original version at https://www.klimaathelpdesk.org/answers/hoe-milieuonvriendelijk-is-vliegen-tegenover-trein-of-bus-met-aanleg-en-onderhoud-van-infrastructuur-meegerekend/, which was published with a CC BY-NC-SA 4.0 license.