# Peer review of "The (non)effect of personalization in climate texts on credibility of climate scientists: A case study on sustainable travel"

_EGUsphere, 2024_

## Author Comment (AC2)

Below, we respond (in black) to the comments by reviewer 2 (in blue):

In this manuscript, the authors examine how personalizing a science text about travel modes and climate change influences participants' attitudes towards climate change and their perception of the writer's credibility. The study reveals that personalization had some effect on participants' emotions but did not significantly alter their attitudes or perceptions of the scientist's credibility. Overall, I found the manuscript very engaging and a thoughtful contribution to Geoscience Communication and the literature. However, adding more nuances to the results and discussion sections would enhance its depth. Below are my comments, which I hope will be helpful to revise your manuscript for publication.

We thank the reviewer for this positive assessment and clear summary of our work. We agree that these comments will help us enhance its clarity and impact.

General comments:

1. It would be beneficial to expand the discussion on scientists expressing their opinions. While you introduce this concept on lines 58-70, revisiting it in the discussion with additional insights would be valuable. Often, scientists are guided by their institutions to avoid taking official stances. Given the urgency of the climate crisis, should scientists feel more comfortable advocating, and should we train future scientists to do this more effectively? Additionally, should workplaces support this type of communication, or should it be left to climate communication experts?

   This is a major topic that the reviewer addresses here; one that would warrant its own article/essay. In fact, we are currently working on such a manuscript. We would thus prefer to leave the deeper discussion on the moral evaluation of advocacy out of this specific manuscript.

2. Why was the personalization conducted in two steps instead of comparing a single personalized version to the original? This approach would have increased the sample sizes for each group. Please provide an explanation in the manuscript to clarify the rationale behind this methodology.

   The reviewer is right that in hindsight we could have used only one changed version of the text. However, when we set out to study the effects, we didn't expect them to be so small. We created two versions because we wanted to explore whether second-person voice was already sufficient to create an effect. In the revised manuscript, we will clarify that by adding "[…] as we aimed to separate the potential effect of the second-person voice from the first-person voice" to the first sentence of section 2.2.

3. Language use, including expressions and humour, is highly dependent on cultural context. A personalized text effective in one language would require careful translation to capture these nuances for different populations, whereas a more

expository text might not need the same level of adjustment. Could you discuss this consideration in the manuscript?

Based also on the comments by reviewer 1, we will highlight in the revised manuscript that "Dutch is very similar to English (they share linguistic roots and numerous similarities in vocabulary, grammar, and syntax), we expect that our results are generalizable to English too".

4. I couldn't find the appendix.

We indeed missed uploading the appendix files. We will do so in the revised submission.

5. Much of the content in section 2.4 seems more appropriate for the results section. Please consider moving this material to a sub-section of the results.

We had indeed considered that but feel that this part about the participants background fits better with the methods section. We therefore prefer to keep it here.

6. The results section could benefit from more detailed description in places. Firstly, some results indicate statistical significance between groups without specifying the direction of the effect. For example, on lines 211-213, we know that there is a difference in responses regarding the writer's perceived credibility, but it is unclear how prior likelihood to fly or science capital and trust (SCT) influence these responses. Similarly, this applies to lines 184-186 and 234-236 (please review for other instances). Secondly, even when results are not statistically significant, I think that it would still be useful to describe them qualitatively, clearly distinguishing between statistically significant and non-significant findings. For instance, regarding SCT on responses to control questions (Fig. 3), although not statistically significant, there is a qualitative difference where a higher percentage of participants with higher SCT found the texts professional/formal. Despite the sample size limiting the detection of small differences as you mention in the discussion, these can still be qualitatively appreciated. Adding such qualitative descriptions throughout the manuscript would highlight nuances not currently captured in the text but shown in the figures. Please consider discussing such subtleties, future research directions, and hypotheses for further studies with larger samples to explore these minimal effects more thoroughly.

This is a good comment. In the revised manuscript, we will add the direction of these statistical differences in the text (although they mostly are also visible in the figures). We prefer not to discuss all the non-significant differences because 1) it would make the text very long and cumbersome to read and 2) these can much more readily be assessed in the figures than from a (long) text.

We do agree with the reviewer that the subtle effects could be borne out with a larger sample size and will add a statement about that in the revised manuscript:

"A larger sample size might make some of the more subtle, non-significant details in the Figures more pronounced."

7. I find the question in section 3.3 about whether the text aims to persuade or inform intriguing, but it should be introduced earlier in the manuscript for better context. Additionally, the impact of participants' SCT on their perception of the writer's impartiality (second question) is not discussed, warranting further exploration in the discussion section.

We prefer to discuss the questions in the order in which we exposed them to the participants, so like to leave it where it is in the results section. In the revised discussion section, we will add the sentence "Participants with high Science Capital and Trust more strongly perceived the goal of the writer to provide impartial information; compared to participants with lower SCT."

8. Participants' prior knowledge (and possibly trust) of the KH platform and the writer's name could have influenced the results. Was this information provided to participants before they answered the questions? Clarifying this could help understand its impact.

We have not checked whether participants had prior knowledge and trust of the KlimaatHelpdesk platform but can expect that only very few of them had visited it before (the KlimaatHelpdesk is not yet that well known). The author of the text is even less known in the Netherlands, so we don't expect there to be a prior trust issue there. In the revised manuscript, we will highlight this in section 2.3.

Specific comments:

9. L18: It would be useful to provide a brief description of the word "attitude" in this context as it is very broad, perhaps using the text in between parentheses on L76.

This is a good point; we will add "(specifically interest and opinion)" to the revised manuscript here.

10. L20-22: This sentence talks about strengthening the link between climate information and action. There needs to be another sentence before that to discuss the existence of that link. Could you cite some literature that discusses the impact of (climate) information for changing attitudes and possibly actions?

This link was (implicitly) inferred in that same sentence, but we will make it explicit in the revised manuscript: "[...] Dong *et al* [2018] found that there is a positive relationship between climate information and action, and that it can be strengthened by [...]"

11. L25-31: I find that this more methodological piece would fit better later in the introduction, once the research questions have been introduced. Where it stands, it disrupts a bit the flow of a very interesting and broader introduction.

This is a good point by the reviewer. We will move this paragraph on IMPACTLAB to after the research questions in the revised manuscript.

12. L32: There is no section 1.1.

We will remove the subheader to section 1.2 in the revised manuscript, so that the introduction is one continuous section.

13. L50-51: You already mentioned that attitudes affect climate action in the first sentences of the introduction. Perhaps rephrase this sentence slightly to incorporate it with the previous sentence and remove the repetition.

This is a good point by the reviewer. We will remove this sentence in the revised manuscript.

14. L99-100: Could you include the full second sentence of the text so that it is a bit easier to interpret the meaning and changes made? Same for conditions 2 and 3.

In the revised manuscript, we will add the rest of this sentence: "[...], because NS and Deutsche Bahn operate mostly on wind energy, and the Swiss railways on hydropower."

15. L116: Did all participants live in The Netherlands?

We can't be certain as we didn't ask that and don't have IP addresses; but given that the survey was in Dutch and distribution via a Dutch website, we assume so.

16. L117-118: Why was the focus on young adults specifically? Please clarify this choice in the manuscript.

In the revised manuscript, we will better clarify that young adults are the target audience of the KlimaatHelpdesk.

17. L121: Does "M" in the parentheses refer to mean or median? This could be written out for clarity and to match the format in section 2.4.1.

Indeed, M refers to Median. We will write this out in the revised version

18. L122: What are the results from the educational level? It is only mentioned in the discussion.

We will add in the revised manuscript that more than 50% of the participants finished higher education.

19. L123: Will the questions be available in the appendix?

All questions are literally in the figures, so we don't think we need to also provide them in the appendix.

20. L127: This is the first time that the word "behaviour" is used in the manuscript. It would be useful to have a brief description of what you mean, and how it situates itself within the attitude, opinion, and action pieces you already mentioned earlier.

Behaviour in this case is simply what they have done before (how many holidays, and how many of these were by plane), but we understand this is confusing. We will therefore change "behaviour" to "conduct" in the revised manuscript.

21. L129-130: Remind us here that the trip from The Netherlands to Milan is the example from the KH text you use - I had forgotten it.

In the revised manuscript, we will add at the end of the sentence that this was the topic of the KlimaatHelpdesk text used in this study.

22. Fig. 1: Could you explain the format of the upper panel in the text and/or in the figure caption so it is easier to understand and interpret this and future figures? It took me a little while to understand at first as I am not used to this type of graphic, with the zero in the middle splitting positive and negative responses.

In the revised version we will add to the caption that "The upper panel uses the plot_likert python package to visualise the number of participants that have given each of the five respective answers; centered around the Neutral (Likert-score=3) value."

23. L151: Could you give a brief description of what the Cronbach-Alpha score measures and its range, so we can interpret the alpha value?

Following also the comment from reviewer 1, we will add that Cronbach-Alpha measures the internal consistency of these six statements and added a reference to Heo et al [2015].

24. L154: You introduced the concept of "effect" in the introduction but I think it would be useful to write it out as "change in attitude" or something similar that

is a bit more descriptive so we easily understand. Same for the title of section 3.1, which could be a bit more descriptive.

We will change this to "difference for" in the text in section 2.4.2 and change the title of section 3,1 to "Change in attitude."

25. Fig. 2: Add the corresponding values for each Likert scale statement in the legend so we know what a score of 4/5 means, and perhaps clarify in the text. I assume 4 corresponds to "agree", but it would be good to be sure.

We will add the Likert scale values to the legend of Figure 2.

26. Fig. 3: Please remind us in the caption that SCT above/equal or below 4 refer to participants with high or low SCT respectively. Same comment for Fig. 4-8.

The caption to these figures also mentions that "score on Science Capital and Trust (right column; higher or lower than 4 out of 5)"; we think this is clear enough.

27. L195-199: I'm curious, did you also look at the effect of prior likelihood to travel by train vs. plane on changes in the participants' opinions on flying or train travel?

We had indeed investigated that but found no difference in whether the participant's opinion had changed between those that were likely to take the plane and those who weren't. We will add a comment about that in the revised section 3.1.

28. Fig. 5: I think you could remove 3 out of the 4 legends as they are all the same.

We will remove the legends from all rows but the first one, to reduce clutter.

29. L213: Please specify that the results separated by likelihood to take the plane are not shown.

We will add to the revised manuscript that this is not shown to the text

30. L250-251: This should be mentioned in the results section as well. Same for L271-272.

We will add statements that "more than 60% of the participants experienced the texts as relatively formal and professional (score <6)" to section 2.5 and "the strongest positive response was on emotion, with >60% of the participants finding the texts interesting" to section 3.1.

31. L254-255: It could also be an additional explanation, and not necessarily an alternative.

We will change "Alternatively" to "Additionally"

32. L259-260: It did change the opinions on flying vs. train travel of some of the participants. Please rephrase or mention the subtlety as per general comment 6.

We will rephrase this to "reading the texts did not change most of the participant's opinions on flying or train travel,"

33. L275: Could you remind us here what RQ2 is, as you do for RQ1 above.

We will add the second research question here

34. L279-280: This is also what you showed with Fig. 2.

We will add a reference to Figure 2 here

35. L299-302: Please mention the figure in the methods section as well.

We will add a comment that all three versions included a fairly technical figure in the first paragraph of section 2.2.

36. L308-309: This is not really what you show though. Please rephrase.

We will remove the sentence that "in general, one text exposure cannot be expected to result in any major shifts in opinion on travel either by plane or train", as we indeed did not explicitly test for that.

Technical corrections:

37. L39-41: This sentence needs rephrasing.

We will rephrase the start of this sentence to "One such element is personalisation …"

38. L52: I would personally prefer using words like "climate mitigation and adaption" instead of "the fight against climate change", but I understand that this is a personal choice.

We will rephrase this indeed to climate mitigation and adaption

39. L182: Remove "were" or add "that" before it.

We will remove "were"

40. L212: "significantly differently". Please check for other instances in the manuscript.

We will change to "differently" here and on other occasions

41. L212-213: "very" is missing in front of "intelligent" and "trustworthy".

We will fix this

42. L234: Should "strongly" be "fully", as per the Fig. 8 legend?

We will change this to Fully now

43. L271-272: Rephrase to "most participants (>60% for all three conditions) found the text interesting, felt calm, [...], and annoyed". Otherwise it's not clear that the >60% applies to all these emotions.

We will change this in the revised version

44. L277: Missing "of" before "sensitivity".

We will fix this

45. L279: Missing "a" before "reflection".

We will fix this

46. L281-282: Missing "perceived" before "more personal".

We will fix this

**References**

Dong Y, S Hu and J Zhu (2018) From source credibility to risk perception: How and when climate information matters to action. *Resources, Conservation and Recycling*, *136*, 410–7

Heo M, N Kim and MS Faith (2015) Statistical power as a function of Cronbach alpha of instrument questionnaire items. *BMC Med Res Methodol*, *15*, 86

---

## Author Response (AR1)

Responses (in black) to the comments by **reviewer 1** (in blue):

The manuscript presents an interesting and relevant subject that is a good fit for *Geoscience Communication*. The study explores the effects of personalization in climate communication, specifically looking at the credibility of climate scientists. Overall, the manuscript is well-structured and easy to read. However, there are several areas that require further development to enhance the clarity and impact of the work.

We thank the reviewer for this very positive assessment and clear summary of our work. We agree that these comments will help us enhance its clarity and impact.

Major comments

1. **Introduction expansion.** The introduction needs expanding, particularly around the IMPACTLAB instrument. It is currently unclear what this instrument is, how it is implemented, and its relevant strengths and weaknesses compared to other approaches. This additional detail will provide a stronger foundation for your study.

   In the revised version, we more clearly explained that the *IMPACTLAB* instrument is a toolbox specifically designed for science communication, that provides a set of tools to measure the effect of public engagement activities. It also includes a decision tree to choose the most appropriate measurement tool for a particular activity. It's based on a theoretical framework to measure three features that help evaluate science communication interventions: science capital (what Peeters *et al* [2022] term "output"), emotional memory ("outcome") and long-term effect ("impact"). Within the framework, it is realized that measuring output is relatively straightforward, but that measuring impact can be extremely difficult. The strength of the tool is that it is very practical and easy to adapt to a wide variety of public engagement activities (lines 75-84 of the track-changed pdf).

2. **Section 1.2 on personalization.** While Section 1.2 on Personalization is good, it would benefit from a more thorough discussion of how other publics, such as lobbyists and politicians, might use this rhetorical technique to undermine the work of scientists. Additionally, exploring how scientists and science communicators can learn from this approach would be valuable. Including examples and references where this approach has worked effectively (or not) in contexts like vaccinations and anti-smoking campaigns would strengthen this section.

   The introduction funnels towards the target audience of the KlimaatHelpdesk, so we don't want to broaden it too much with other audiences etc. However, we now mention that lobbyists use manipulation of narrative elements in the revised manuscript (line 36 of the track-changed pdf). We have also searched for examples and references where personalization of scientific texts has been studied, but couldn't find much.

3. **Selection of KH article.** Clarify how the KH article was chosen in terms of topic. Discuss the implications of this choice for the results. For instance, consider whether

topics such as ocean acidification or sea level rises affecting Indigenous communities might have yielded different outcomes.

The reviewer has a good point that the motivation for choosing this particular text was missing from the original manuscript. In the revised manuscript, we explained that we chose this text on the climate impact of train versus plane travel because it had received a big readership on the website, so we knew it was a popular topic, was a relatively short text, and because it was not too technical and therefore relatively easy to adapt (lines 107-109 of the track-changed pdf).

4. **Translation and personalization.** The issue of translation requires further exploration. Discuss how personalization might differ in Dutch compared to English, and what this implies for other languages. This will help in understanding the broader applicability of your findings.

    In the revised manuscript, we highlighted that Dutch is very similar to English (they share linguistic roots and numerous similarities in vocabulary, grammar, and syntax), so we expect that our results are generalizable to English too (lines 132-134 of the track-changed pdf).

5. **SurveySwap details.** Provide more information about *SurveySwap*. Explain what it is and how it works in terms of reliability. This will help readers assess the robustness of your methodology.

    In the revised manuscript, we explained that SurveySwap is an online platform that operates on a reciprocal basis where users can earn credits by completing other users' surveys, and then use those credits to have their own surveys completed. This system is particularly used by students and academics who need to collect a significant amount of data for their research projects or dissertations, but the pool of respondents may therefore be limited in diversity (lines 136-140 of the track-changed pdf).

6. **Consent forms and ethical considerations.** Expand on the discussion around consent forms. What risks and benefits did you consider and communicate to participants? How was the data stored securely? Include more details on these aspects. Additionally, bring some of the 'Ethical statement' (line 325) into the main body of the text, such as the classification of the study as low risk, which negated the need for further ethical review or privacy assessment.

    This is a good idea. We added a few sentences on the topic of privacy and ethics to the beginning of section 2.3 in the revised manuscript (lines 141-144 of the track-changed pdf).

7. **Determining personalization levels.** Explain how you determined if a text was slightly or highly personalized. Acknowledge the potential subjectivity here and discuss how this assessment could be repeated reliably.

The level of personalization was not determined *a posteriori*, but was defined a priori in our research setup. As discussed in section 2.2, 'slightly personalized' means changing definite articles and indefinite pronouns to second-person (possessive) pronouns. In the 'highly personalized' texts, we furthermore added the first-person voice of the writer. Indeed, some of this might be subjective, but the results in Figure 3 suggest that indeed the highly personalised text was perceived as the most personal one.

In the revised manuscript, we clarified this further, by repeating this difference at the start of the discussion section (lines 295-296 of the track-changed pdf).

8. **Title scope.** The study focuses on one example – sustainable travel. The current title, "The (non)effect of personalization in climate texts on the credibility of climate scientists," is too broad. Consider revising the title to better reflect the scope of the study, such as "The (non)effect of personalization in climate texts on the credibility of climate scientists: A case study on sustainable travel."

This is a good suggestion; we changed the title in the revised version of the manuscript.

9. The conclusions currently read more like a summary of the article. Expand this section to include the significance of the work and its implications for future studies. This will provide a more impactful closing to your manuscript.

This is also a good suggestion. In the revised manuscript, we added that this is only one study on one text with one type of audience (Dutch young adults). If our results hold up in a wider variety of texts and audiences, this suggests that adding personalization does not harm the message in climate communication materials, which is a useful finding for communication professionals who aim to make climate texts more engaging (lines 367-370 of the track-changed pdf).

**Minor Comments**

1. **Abstract.** The abstract is well-crafted and easy to digest. No changes are necessary here.

We are happy to hear this.

2. **Figures and captions.** The figures are clear, well-produced, and effectively support the text. However, the captions could benefit from a little more detail so they can be understood independently of the main body of the article.

In the revised version, we ended each caption with a one-sentence take-home message of the figure, to indeed help readers that aim to understand the point of the paper from the abstract and figures alone.

3. **Section 2.5 on quantitative statistics.** Section 2.5 on quantitative statistics for the control is sound and well-explained but does not integrate smoothly with the

preceding and following sections. Improving the transitions will enhance the overall flow of the manuscript.

We now understand that 'Control' may be too vague a term, so we changed the title of section 2.5 to "Assessment of differences between conditions", and reworded this section to make it more in line with the rest of the manuscript (lines 196-197 of the track-changed pdf).

4. **Statistical tests explanation.** Every time you introduce a statistical test or method, such as the Cronbach-Alpha score, provide an explanation of what the test is and what certain scores mean. This will help the reader understand the methodology better and make the manuscript more accessible.

We added explanations and references to the literature for statistical tests such as Cronbach's alpha and Holm correction (lines 183-184 and 201 of the track-changed pdf).

5. **Discussion and Research Questions.** The discussion would benefit from clearer presentation of the Research Questions and their integration throughout the text. This will help in drawing direct connections between your findings and the questions posed.

While we do mention both Research Questions in the Discussion, we realise that this might not be too easy to find. In the revised manuscript, we therefore repeated the verbatim research questions in the discussion and highlight them through italics font (lines 305-306 and 326-327 of the track-changed pdf).

6. **Fair discussion of limitations.** The discussion of limitations is very fair and well done. This transparency strengthens the credibility of your study.

We thank the reviewer for this supportive comment.

Responses (in black) to the comments by **reviewer 2** (in blue):

In this manuscript, the authors examine how personalizing a science text about travel modes and climate change influences participants' attitudes towards climate change and their perception of the writer's credibility. The study reveals that personalization had some effect on participants' emotions but did not significantly alter their attitudes or perceptions of the scientist's credibility. Overall, I found the manuscript very engaging and a thoughtful contribution to Geoscience Communication and the literature. However, adding more nuances to the results and discussion sections would enhance its depth. Below are my comments, which I hope will be helpful to revise your manuscript for publication.

We thank the reviewer for this positive assessment and clear summary of our work. We agree that these comments will help us enhance its clarity and impact.

General comments:

1. It would be beneficial to expand the discussion on scientists expressing their opinions. While you introduce this concept on lines 58-70, revisiting it in the discussion with additional insights would be valuable. Often, scientists are guided by their institutions to avoid taking official stances. Given the urgency of the climate crisis, should scientists feel more comfortable advocating, and should we train future scientists to do this more effectively? Additionally, should workplaces support this type of communication, or should it be left to climate communication experts?

   This is a major topic that the reviewer addresses here; one that would warrant its own article/essay. In fact, we are currently working on such a manuscript. We would thus prefer to leave the deeper discussion on the moral evaluation of advocacy out of this specific manuscript.

2. Why was the personalization conducted in two steps instead of comparing a single personalized version to the original? This approach would have increased the sample sizes for each group. Please provide an explanation in the manuscript to clarify the rationale behind this methodology.

   The reviewer is right that in hindsight we could have used only one changed version of the text. However, when we set out to study the effects, we didn't expect them to be so small. We created two versions because we wanted to explore whether second-person voice was already sufficient to create an effect. In the revised manuscript, we clarified that by adding "[…] as we aimed to separate the potential effect of the second-person voice from the first-person voice" to the first sentence of section 2.2 (line 106-107 of the track-changed pdf).

3. Language use, including expressions and humour, is highly dependent on cultural context. A personalized text effective in one language would require careful translation to capture these nuances for different populations, whereas a more

expository text might not need the same level of adjustment. Could you discuss this consideration in the manuscript?

Based also on the comments by reviewer 1, we highlighted in the revised manuscript that "Dutch is very similar to English (they share linguistic roots and numerous similarities in vocabulary, grammar, and syntax), we expect that our results are generalizable to English too" (lines 132-134 of the track-changed pdf).

4. I couldn't find the appendix.

   We indeed missed uploading the appendix files. We did so in the revised submission (pages 19-21 of the track-changed pdf).

5. Much of the content in section 2.4 seems more appropriate for the results section. Please consider moving this material to a sub-section of the results.

   We had indeed considered that but feel that this part about the participants background fits better with the methods section. We therefore prefer to keep it here.

6. The results section could benefit from more detailed description in places. Firstly, some results indicate statistical significance between groups without specifying the direction of the effect. For example, on lines 211-213, we know that there is a difference in responses regarding the writer's perceived credibility, but it is unclear how prior likelihood to fly or science capital and trust (SCT) influence these responses. Similarly, this applies to lines 184-186 and 234-236 (please review for other instances). Secondly, even when results are not statistically significant, I think that it would still be useful to describe them qualitatively, clearly distinguishing between statistically significant and non-significant findings. For instance, regarding SCT on responses to control questions (Fig. 3), although not statistically significant, there is a qualitative difference where a higher percentage of participants with higher SCT found the texts professional/formal. Despite the sample size limiting the detection of small differences as you mention in the discussion, these can still be qualitatively appreciated. Adding such qualitative descriptions throughout the manuscript would highlight nuances not currently captured in the text but shown in the figures. Please consider discussing such subtleties, future research directions, and hypotheses for further studies with larger samples to explore these minimal effects more thoroughly.

   This is a good comment. In the revised manuscript, we added the direction of these statistical differences in the text (although they mostly are also visible in the figures). We prefer not to discuss all the non-significant differences because 1) it would make the text very long and cumbersome to read and 2) these can much more readily be assessed in the figures than from a (long) text.

   We do agree with the reviewer that the subtle effects could be borne out with a larger sample size and added a statement about that in the revised manuscript:

"A larger sample size might make some of the more subtle, non-significant details in the Figures more pronounced." (lines 347-348 of the track-changed pdf).

7. I find the question in section 3.3 about whether the text aims to persuade or inform intriguing, but it should be introduced earlier in the manuscript for better context. Additionally, the impact of participants' SCT on their perception of the writer's impartiality (second question) is not discussed, warranting further exploration in the discussion section.

We prefer to discuss the questions in the order in which we exposed them to the participants, so like to leave it where it is in the results section. In the revised discussion section, we added the sentence "Participants with high Science Capital and Trust more strongly perceived the goal of the writer to provide impartial information; compared to participants with lower SCT." (lines 332-334 of the track-changed pdf).

8. Participants' prior knowledge (and possibly trust) of the KH platform and the writer's name could have influenced the results. Was this information provided to participants before they answered the questions? Clarifying this could help understand its impact.

We have not checked whether participants had prior knowledge and trust of the KlimaatHelpdesk platform but can expect that only very few of them had visited it before (the KlimaatHelpdesk is not yet that well known). The author of the text is even less known in the Netherlands, so we don't expect there to be a prior trust issue there. In the revised manuscript, we highlighted this in section 2.3 (lines 149-150 of the track-changed pdf).

Specific comments:

9. L18: It would be useful to provide a brief description of the word "attitude" in this context as it is very broad, perhaps using the text in between parentheses on L76.

This is a good point; we added "(specifically interest and opinion)" to the revised manuscript (line 18 of the track-changed pdf).

10. L20-22: This sentence talks about strengthening the link between climate information and action. There needs to be another sentence before that to discuss the existence of that link. Could you cite some literature that discusses the impact of (climate) information for changing attitudes and possibly actions?

This link was (implicitly) inferred in that same sentence, but we made it explicit in the revised manuscript: "[...] Dong *et al* [2018] found that there is a positive

relationship between climate information and action, and that it can be strengthened by [...]" (line 21-22 of the track-changed pdf).

11. L25-31: I find that this more methodological piece would fit better later in the introduction, once the research questions have been introduced. Where it stands, it disrupts a bit the flow of a very interesting and broader introduction.

This is a good point by the reviewer. We moved this paragraph on IMPACTLAB to after the research questions in the revised manuscript (lines 74-84 of the track-changed pdf).

12. L32: There is no section 1.1.

We removed the subheader to section 1.2 in the revised manuscript, so that the introduction is one continuous section.

13. L50-51: You already mentioned that attitudes affect climate action in the first sentences of the introduction. Perhaps rephrase this sentence slightly to incorporate it with the previous sentence and remove the repetition.

This is a good point by the reviewer. We removed this sentence in the revised manuscript.

14. L99-100: Could you include the full second sentence of the text so that it is a bit easier to interpret the meaning and changes made? Same for conditions 2 and 3.

In the revised manuscript, we added the rest of this sentence: "[...], because NS and Deutsche Bahn operate mostly on wind energy, and the Swiss railways on hydropower." (lines 115-116 of the track-changed pdf)

15. L116: Did all participants live in The Netherlands?

We can't be certain as we didn't ask that and don't have IP addresses; but given that the survey was in Dutch and distribution via a Dutch website, we assume so.

16. L117-118: Why was the focus on young adults specifically? Please clarify this choice in the manuscript.

In the revised manuscript, we better clarified that young adults are the target audience of the KlimaatHelpdesk (line 103 of the track-changed pdf).

17. L121: Does "M" in the parentheses refer to mean or median? This could be written out for clarity and to match the format in section 2.4.1.

Indeed, M refers to Median. We have now written this out in the revised version

18. L122: What are the results from the educational level? It is only mentioned in the discussion.

We added in the revised manuscript that more than 50% of the participants finished higher education (line 152 of the track-changed pdf).

19. L123: Will the questions be available in the appendix?

All questions are literally in the figures, so we don't think we need to also provide them in the appendix.

20. L127: This is the first time that the word "behaviour" is used in the manuscript. It would be useful to have a brief description of what you mean, and how it situates itself within the attitude, opinion, and action pieces you already mentioned earlier.

Behaviour in this case is simply what they have done before (how many holidays, and how many of these were by plane), but we understand this is confusing. We therefore changed "behaviour" to "conduct" in the revised manuscript (e.g. line 156 of the track-changed pdf).

21. L129-130: Remind us here that the trip from The Netherlands to Milan is the example from the KH text you use - I had forgotten it.

In the revised manuscript, we added at the end of the sentence that this was the topic of the KlimaatHelpdesk text used in this study (line 161 of the track-changed pdf).

22. Fig. 1: Could you explain the format of the upper panel in the text and/or in the figure caption so it is easier to understand and interpret this and future figures? It took me a little while to understand at first as I am not used to this type of graphic, with the zero in the middle splitting positive and negative responses.

In the revised version we added to the caption that "The upper panel uses the plot_likert python package to visualise the number of participants that have given each of the five respective answers; centered around the Neutral (Likert-score=3) value."

23. L151: Could you give a brief description of what the Cronbach-Alpha score measures and its range, so we can interpret the alpha value?

Following also the comment from reviewer 1, we added that Cronbach-Alpha measures the internal consistency of these six statements (lines 183-184 of the track-changed pdf) and added a reference to Heo et al [2015].

24. L154: You introduced the concept of "effect" in the introduction but I think it would be useful to write it out as "change in attitude" or something similar that is a bit more descriptive so we easily understand. Same for the title of section 3.1, which could be a bit more descriptive.

We changed this to "difference for" in the text in section 2.4.2 (line 187 of the track-changed pdf) and changed the title of section 3,1 to "Change in attitude."

25. Fig. 2: Add the corresponding values for each Likert scale statement in the legend so we know what a score of 4/5 means, and perhaps clarify in the text. I assume 4 corresponds to "agree", but it would be good to be sure.

We added the Likert scale values to the legend of Figure 2.

26. Fig. 3: Please remind us in the caption that SCT above/equal or below 4 refer to participants with high or low SCT respectively. Same comment for Fig. 4-8.

The caption to these figures also mentions that "score on Science Capital and Trust (right column; higher or lower than 4 out of 5)"; we think this is clear enough.

27. L195-199: I'm curious, did you also look at the effect of prior likelihood to travel by train vs. plane on changes in the participants' opinions on flying or train travel?

We had indeed investigated that but found no difference in whether the participant's opinion had changed between those that were likely to take the plane and those who weren't. We added a comment about that in the revised section 3.1 (lines 240-241 of the track-changed pdf).

28. Fig. 5: I think you could remove 3 out of the 4 legends as they are all the same.

We removed the legends from all rows but the first one, to reduce clutter.

29. L213: Please specify that the results separated by likelihood to take the plane are not shown.

We added to the revised manuscript that this is not shown to the text (line 257 of the track-changed pdf)

30. L250-251: This should be mentioned in the results section as well. Same for L271-272.

We added statements that "more than 60% of the participants experienced the texts as relatively formal and professional (score <6)" to section 2.5 (lines 198-199 of the track-changed pdf) and "the strongest positive response was on emotion, with >60% of the participants finding the texts interesting" to section 3.1 (lines 221-222 of the track-changed pdf).

31. L254-255: It could also be an additional explanation, and not necessarily an alternative.

We have changed "Alternatively" to "Additionally" (line 303 of the track-changed pdf)

32. L259-260: It did change the opinions on flying vs. train travel of some of the participants. Please rephrase or mention the subtlety as per general comment 6.

We rephrased this to "reading the texts did not change most of the participant's opinions on flying or train travel," (line 309 of the track-changed pdf)

33. L275: Could you remind us here what RQ2 is, as you do for RQ1 above.

We added the second research question here (line 326-327 of the track-changed pdf)

34. L279-280: This is also what you showed with Fig. 2.

We added a reference to Figure 2 here (line 331 of the track-changed pdf)

35. L299-302: Please mention the figure in the methods section as well.

We added a comment that all three versions included a fairly technical figure in the first paragraph of section 2.2 (line 110 of the track-changed pdf).

36. L308-309: This is not really what you show though. Please rephrase.

We removed the sentence that "in general, one text exposure cannot be expected to result in any major shifts in opinion on travel either by plane or train", as we indeed did not explicitly test for that.

Technical corrections:

37. L39-41: This sentence needs rephrasing.

We rephrased the start of this sentence to "One such element is personalisation …" (line 39 of the track-changed pdf)

38. L52: I would personally prefer using words like "climate mitigation and adaption" instead of "the fight against climate change", but I understand that this is a personal choice.

We rephrased this to climate mitigation and adaption (line 53 of the track-changed pdf)

39. L182: Remove "were" or add "that" before it.

We have removed "were" (line 219 of the track-changed pdf)

40. L212: "significantly differently". Please check for other instances in the manuscript.

We have changed to "differently" here (line 239 of the track-changed pdf) and on other occasions

41. L212-213: "very" is missing in front of "intelligent" and "trustworthy".

We have fixed this on line 256 of the track-changed pdf

42. L234: Should "strongly" be "fully", as per the Fig. 8 legend?

We have changed this to Fully now (lines 268 and 281 of the track-changed pdf)

43. L271-272: Rephrase to "most participants (>60% for all three conditions) found the text interesting, felt calm, […], and annoyed". Otherwise it's not clear that the >60% applies to all these emotions.

We have changed this in the revised version (line 322 of the track-changed pdf)

44. L277: Missing "of" before "sensitivity".

We have fixed this (line 329 of the track-changed pdf)

45. L279: Missing "a" before "reflection".

We have changed the wording (line 332 of the track-changed pdf)

46. L281-282: Missing "perceived" before "more personal".

We have fixed this (line 335 of the track-changed pdf.

Responses (in black) to the comments by **the editor** (in blue):

In addition to the reviewers comments, minor suggestions from myself would be

1.  to briefly define the term 'personalization' more explicitly early on

    We now more explicitly give the definition of personalization on line 40 of the track-changed pdf.

2.  to state the length of the text given to participants.

    In the revised manuscript, we have added the length (approximately 750 words) in line 109-110 of the track-changed pdf

3.  Perhaps there is a minor typo in the abstract (line 10) - 'we exposed [one] hundred participants...'

    We have fixed this type in the abstract (line 10 of the track-changed pdf)

**References**

Dong Y, S Hu and J Zhu (2018) From source credibility to risk perception: How and when climate information matters to action. *Resources, Conservation and Recycling*, *136*, 410–7

Heo M, N Kim and MS Faith (2015) Statistical power as a function of Cronbach alpha of instrument questionnaire items. *BMC Med Res Methodol*, *15*, 86

---

## Author Response (AR2)

Responses (in black) to the comments by **Solmaz Mohadjer** (in blue):

1. Remove informal language (e.g., it's, haven't, weren't etc.

   We have replaced all occurences of it's, haven't, weren't and can't in the track-changed pdf (lines 68, 141, 232, 302), except where the word was part of the questionnaire.

2. Define 'young secondary school students' by providing an age range in parentheses.

   We have now added that the readership is ages 13-35 (line 94 of the track-changed pdf).

3. Line 108: large readership (xx people)

   We have now added that this article had more than 5,000 visits (line 99 of the track-changed pdf)

4. What does NS stand for? Please spell it out.

   We have now changed this to "Nederlandse Spoorwegen" (lines 106, 113 and 122 of the track-changed pdf)

5. Line 115: Deutsche Bahn (national railway company in Germany)

   We have added this to the track-changed pdf.

Responses (in black) to the comments by **Jenna Sutherland** (in blue):

6. Line 28 - Insert the word 'written' - '(specifically [written] in a manipulative way, such as..'

   We have added this word (line 28 in the track-changed pdf)

7. Line 106 - I understand that this quoted text has come from the original source but it is not clear what NS is. I think the acronym needs to be defined. I assume it is Dutch national railway company Nederlandse Spoorwegen

   This is correct, in the revised manuscript we have changed the wording to "[...] because Nederlandse Spoorwegen and Deutsche Bahn (the Dutch and German national railway companies) [...]" (lines 106-123 in the track-changed pdf)

Responses (in black) to the comments by **Lorena Grabowski** (in blue):

8. Please ensure that the colour schemes used in your maps and charts allow readers with colour vision deficiencies to correctly interpret your findings (see e.g. F2). Please check your figures using the Coblis Color Blindness Simulator (https://www.color-blindness.com/coblis-color-blindness-simulator/) and revise the colour schemes accordingly.

   We have confirmed that all our figures can be correctly interpreted by readers with colour vision deficiencies now. We have used hashing in the Figures to differentiate red and green.